Copyright waived. CC0 1.0.

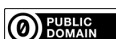



# Reconstructing the dynamics of the highly-similar May 2016 and June 2019 Iliamna Volcano, Alaska ice–rock avalanches from seismoacoustic data

Liam Toney[1], David Fee[1], Kate E. Allstadt[2], Matt Haney[3], and Robin S. Matoza[4]

[1]Alaska Volcano Observatory and Wilson Alaska Technical Center, Geophysical Institute, University of Alaska Fairbanks, Fairbanks, AK, USA
[2]U.S. Geological Survey Geologic Hazards Science Center, Golden, CO, USA
[3]U.S. Geological Survey Alaska Volcano Observatory, Anchorage, AK, USA
[4]Department of Earth Science and Earth Research Institute, University of California, Santa Barbara, CA, USA

**Correspondence:** Liam Toney (ldtoney@alaska.edu)

**Abstract.** Surficial mass wasting events are a hazard worldwide. Seismic and acoustic signals from these often-remote processes, combined with other geophysical observations, can provide key information for monitoring and rapid response efforts and enhance our understanding of event dynamics. Here we present seismoacoustic data and analyses for two very large ice–rock avalanches occurring on the "natural laboratory" of Iliamna Volcano, Alaska (USA) on 22 May 2016 and 21 June 2019.
5  Iliamna is a glacier-mantled stratovolcano located in the Cook Inlet, ∼200 km from Anchorage, Alaska. The volcano experiences massive, quasi-annual slope failures due to glacial instabilities and hydrothermal alteration of material near its summit. The May 2016 and June 2019 avalanches were particularly large and generated energetic seismic and infrasound signals which were recorded on numerous stations at ranges from ∼9 to over 600 km. Both avalanches initiated in the same location near the head of Iliamna's east-facing Red Glacier, and their ∼8 km long runout shapes are nearly identical. This repeatability –
10  which is rare for mass movements – provides an excellent opportunity for comparison and validation of seismoacoustic source characteristics. For both events, we invert long-period (15–80 s) seismic signals to obtain a force-time representation of the source. We model the avalanche as a sliding block which exerts a spatially-static point force on the Earth. We use this force-time function to derive constraints on avalanche acceleration, velocity, and directionality which are compatible with satellite imagery and observed terrain features. Our inversion results suggest that the avalanches reached speeds exceeding 80 m s$^{-1}$,
15  consistent with numerical modeling from previous Iliamna studies. We lack sufficient local infrasound data to test an acoustic source model for these processes. However, the acoustic data suggest that infrasound from these avalanches is produced after the mass movement regime transitions from cohesive block-type failure to granular and turbulent flow – little to no infrasound is generated by the initial failure. At Iliamna, synthesis of advanced numerical flow models and more detailed groundtruth combined with increased geophysical station coverage could yield significant gains in our understanding of these events.

Copyright waived. CC0 1.0.

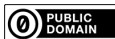



## 1 Introduction

Surficial gravitational mass movements, such as debris flows, rockfalls, lahars, and avalanches, constitute a broad collection of Earth processes which are a significant hazard around the world (Voight, 1978). These events can cause devastating damage to life and property when they occur in at-risk, populated areas in mountainous regions or on the flanks of volcanoes. Avalanches involving mixtures of ice and rock are a subset of these processes usually occurring in topographically extreme, glaciated terrain. Some of the most deadly surficial gravitational mass movements (hereafter, just "mass movements") in history were ice–rock avalanches. For example, the Huascarán avalanches occurring in 1962 and 1970 in the Peruvian Andes together claimed an estimated 22,000 lives (Plafker and Ericksen, 1978). However, due to their high mobility and frequently remote location, eyewitness observations of these dramatic processes are rare (Caplan-Auerbach and Huggel, 2007; Coe et al., 2016), and other assessment methods such as geologic mapping or satellite imagery analysis may not be timely or even possible due to the rugged terrain and volatile mountain weather typically found in such settings

Seismoacoustics is an emerging tool which can help us understand these powerful yet elusive processes (Allstadt et al., 2018, and references therein). Mass movements transfer energy into the solid Earth as seismic waves and into the atmosphere as acoustic waves. The atmospheric waves are primarily in the infrasonic range at frequencies below the range of human hearing (< 20 Hz). These signals contain valuable and complementary information about the character and size of the event, and also provide a high-resolution record of event timing. Even moderately-sized mass movements can be recorded from sufficiently safe distances. By analyzing the seismic and acoustic waves generated by these processes, we can better understand their dynamics and work towards improved hazard mitigation and response. Seismology and infrasound are therefore some of most promising tools for near-real-time detection and characterization of remote mass movements (Allstadt et al., 2018). However, development of detailed seismoacoustic source models is still an area of active research, as relatively few high-resolution recordings exist.

Here, we focus on two ice–rock avalanches occurring in May 2016 and June 2019 on Iliamna Volcano, Alaska, USA. These avalanches were very large, each measuring ∼8 km from crown to toe. Both events produced energetic seismic and acoustic signals broadly recorded at local (< 100 km) and regional (> 100 km) distances. Relatively dense regional seismic and acoustic networks were in place during these events (Fig. 1), providing a unique opportunity for source quantification and comparison. Additionally, the location and nature of failure and the material, shape, and size of the resulting deposits are very similar between the two events (Fig. 2), providing excellent datasets for comparison. Iliamna Volcano is known for frequent, large mass movements of this nature (e.g., Caplan-Auerbach et al., 2004; Caplan-Auerbach and Huggel, 2007; Huggel et al., 2007; Schneider et al., 2010).

Numerous studies have analyzed seismic and acoustic data from mass movements (see Allstadt et al., 2018, and references therein). Here we apply consistent methodology to analyze two very similar events with excellent regional seismoacoustic station coverage. The repeatability of the Iliamna Volcano avalanches facilitates validation of source models and, in the case of infrasound, allows for separation of the acoustic source from typically highly time-dependent atmospheric path effects.

Copyright waived. CC0 1.0.

Earth **Surface**
**Dynamics**
Discussions

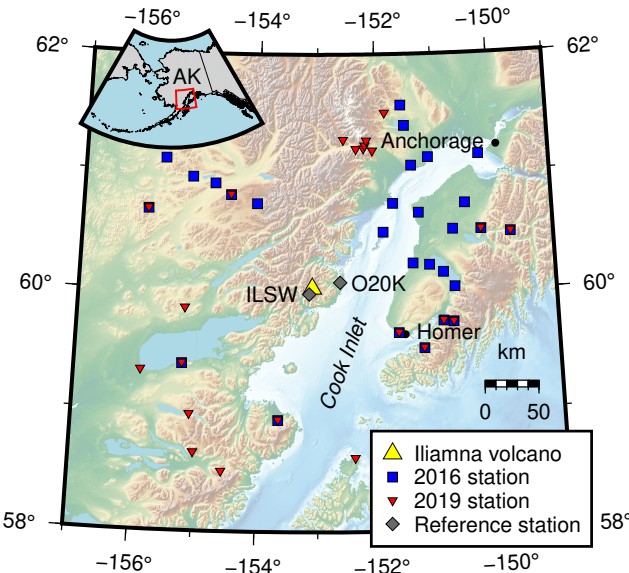

**Figure 1.** Map of the Cook Inlet region, Alaska. Broadband seismic stations used in the 2016 (28 stations) and 2019 (23 stations) force inversions are shown as blue squares and red triangles, respectively. Overlapping markers denote stations used in both inversions. The station distribution varies greatly between the two events due to the presence of a temporary seismic array in 2016 and increased Transportable Array station coverage in 2019. Reference stations ILSW and O20K (the closest seismometer and infrasound sensor to the events, respectively) are shown as gray diamonds. The city of Anchorage and town of Homer are marked as black dots. Red box in inset shows main map extent.

In this study, we describe the acoustic and seismic signals generated by the 2016 and 2019 Iliamna Volcano avalanches, along with auxiliary information including aerial, ground-based, and satellite imagery. We explore the timing and strength

of the avalanche acoustic signal and assess the possibility of acoustic source directionality. We invert the strong long-period seismic signals produced by the events to obtain the time series of forces that the center of mass (COM) of each avalanche exerted on the Earth – the "force-time function". From there, we calculate the acceleration, velocity, and displacement of the COMs and compare these to auxiliary data such as digital elevation models and satellite imagery. Our modeled forces and trajectories generally agree well with the satellite imagery and observed terrain features and offer insight into the acoustic

source properties of these massive avalanches.

## 2 Background

### 2.1 Analysis of long-period seismic waves from mass movements

The amplitude and frequency content of the seismic wavefield radiated by a surficial mass movement are strongly controlled by the spatial and temporal scales involved as well as the structural coherence of the moving material. Processes such as powdery

snow avalanches and lahars, which primarily involve incoherent collections of fine-grained particles, produce relatively high-

Copyright waived. CC0 1.0.

Earth **Surface**
**Dynamics**
Discussions

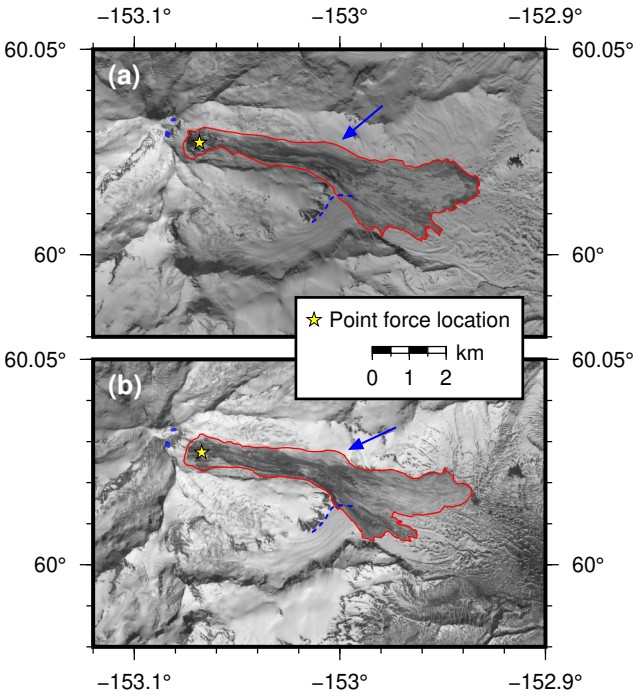

**Figure 2.** Satellite images of the 2016 and 2019 Red Glacier avalanche deposits acquired on **(a)** 23 May 2016 and **(b)** 22 June 2019, both less than 48 h post-event. Red outlines delineate approximate avalanche extents (source, track, and deposit areas). Yellow stars mark the location of the inversion point force. Blue arrows indicate the location of superelevation-like flow lobes. Blue dashed lines delineate the northern margin of an unnamed tributary glacier which joins Red Glacier from the southwest. Translucent blue patches show the approximate locations of two fumarole zones located to the east of the summit. Imagery © 2016 and 2019 Planet Labs, Inc.

frequency seismicity (Allstadt et al., 2018). However, larger events which move coherently – such as rockfalls and ice–rock avalanches – can additionally produce significant long-period (> 10 s) seismic energy that can be recorded globally (Hibert et al., 2017; Allstadt et al., 2018). These long-period seismic waves originate from the bulk acceleration and deceleration of mass as it moves downslope (Ekström and Stark, 2013).

Long-period seismic waves can be used to invert for quantitative mass movement source properties. The wave propagation (i.e. Green's function) at these periods is often straightforward to model due to the relatively small influence of topography and Earth structure on such long-wavelength signals. Once the propagation is accounted for, one can invert for the time-varying force vector that the moving mass exerted on the Earth (e.g., Kawakatsu, 1989; Allstadt, 2013; Ekström and Stark, 2013; Coe et al., 2016; Gualtieri and Ekström, 2018). The trajectory can then be obtained if the mass, generally assumed to be

constant, is known or can be estimated (e.g., Ekström and Stark, 2013; Moore et al., 2017; Gualtieri and Ekström, 2018; Schöpa et al., 2018). However, complexities such as entrainment and deposition along the flow path clearly violate the constant mass approximation, so this method has generally only been successful for simple runout paths. The infrequent nature of catastrophic

Copyright waived. CC0 1.0.

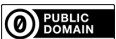



mass movements capable of generating sufficiently long-period seismic radiation means that opportunities to apply this model are limited (Hibert et al., 2017).

**2.2 Acoustic studies of mass movements**

More recently, studies have incorporated observations and analysis of infrasound generated by mass movements. Since infrasound is often deployed in a volcano-monitoring context (Fee and Matoza, 2013; Matoza et al., 2019), many acoustic observations of mass movements have documented volcanic phenomena such as pyroclastic flows (e.g., Yamasato, 1997; Ripepe et al., 2009, 2010; Delle Donne et al., 2014), lahars (e.g., Johnson and Palma, 2015), rockfalls (e.g., Moran et al., 2008; Johnson and 85 Ronan, 2015), and flank collapse events (e.g., Perttu et al., 2020). Outside of the volcanic context, debris flows (e.g., Kogelnig et al., 2014; Schimmel and Hübl, 2016; Marchetti et al., 2019b), powder snow avalanches, (e.g., Ulivieri et al., 2011; Havens et al., 2014; Marchetti et al., 2015, 2019a), non-volcanic rockfalls (e.g., Zimmer et al., 2012; Zimmer and Sitar, 2015), and rock avalanches (e.g., Moore et al., 2017) have been observed acoustically. Infrasound recordings of large surficial mass flows are rare, particularly at local and regional distances.

Infrasonic source directionality has previously been assessed for dense recordings of volcanic explosions. For example, Iezzi et al. (2019) performed a multipole acoustic source inversion on explosions from Yasur volcano, Vanuatu, describing the source as a combination of monopole (uniform radiation) and dipole (directional radiation) components. Mass movement acoustic radiation has been suggested to be highly directional and potentially described by an acoustic dipole (Haney et al., 2018; Allstadt et al., 2018). However, assessment of source directionality for mass movements requires dense station coverage 95 which is not usually available; therefore the actual source directionality has not been validated with data. Additionally, beyond local distances, path effects from the usually highly spatiotemporally variable atmosphere become important. These effects can mask source directionality or produce spurious source directionality and must be accounted for (e.g., Perttu et al., 2020).

Arrays of infrasound sensors can be used to determine the backazimuth of incident acoustic waves and can track flow fronts in certain circumstances (e.g., Johnson and Palma, 2015; Marchetti et al., 2019a). Though infrasonic records of mass movements 100 are becoming more common, the acoustic source theory is currently underdeveloped (Allstadt et al., 2018). Very simple mass movements such as rockfalls have been treated as monopoles (e.g., Moran et al., 2008), but often the source of infrasound is moving and distributed, complicating modeling. Marchetti et al. (2019b) modeled a debris flow as a linear series of monopole sources in motion, but found that infrasound array processing results always pointed back to fixed locations corresponding to check dams in the debris flow drainage, the most acoustically energetic sources. Using infrasound arrays, Johnson and Palma 105 (2015) tracked a lahar which registered as a moving source until it encountered a topographic notch, at which point the source location became fixed on this acoustically "loud" flow feature. The dynamic, spatiotemporal variability of the atmosphere also complicates infrasound source modeling (Poppeliers et al., 2020). These studies highlight the challenge in determining the source of mass movement generated infrasound.

Copyright waived. CC0 1.0.



### 2.3  Ice–rock avalanches

Ice–rock avalanches are a subset of mass movements which consist of rapid flows of pulverized ice and rock. They are characterized by their exceptionally high mobility and often catastrophic size (Schneider et al., 2011; Hungr et al., 2014). Though the initial failure of an ice–rock avalanche can free larger blocks of material, such blocks quickly disintegrate into small fragments of rock and ice as they impact asperities in the flow path at speed. This debris travels on a saturated layer of material which essentially acts as a lubricated bed surface, increasing mobility (Hungr et al., 2014). Additionally, since these processes often

take place in steep, heavily glaciated terrain (Schneider et al., 2011), the avalanches commonly flow over glaciers. This further enhances mobility due to the low friction of glacier ice (Schneider et al., 2010). Owing to their high mobility and often large volumes, debris avalanches such as ice–rock avalanches are among the most seismogenic types of mass movements (Allstadt et al., 2018).

### 2.4  Iliamna Volcano, Alaska

Iliamna Volcano (hereafter, "Iliamna") is a 3,053-meter-tall stratovolcano located in the Cook Inlet region of south-central Alaska, USA (Fig. 1). The volcano lies about 215 km from the city of Anchorage, and roughly 100 km across the Cook Inlet from the town of Homer. The geology of Iliamna consists primarily of stratified andesitic lava flows with smaller contributions from mass wasting deposits of various types. The volcano's summit is perennially mantled with snow and ice, and its edifice hosts several large valley glaciers (Waythomas and Miller, 1999). Two zones of sulfurous fumaroles located on the eastern side

of Iliamna's summit (see blue translucent patches in Fig. 2) emit steam and volcanic gas quasi-continuously (Werner et al., 2011).

Though Iliamna has not erupted in historical time, it experienced two periods of seismic unrest occurring in 1996 and 2012 which were interpreted as magmatic intrusions and failed eruptions (Roman et al., 2004; Herrick et al., 2014). Additionally, the deeply dissected and hydrothermally altered edifice of Iliamna hosts frequent mass wasting events. Geologic evidence of late

Holocene lahars and debris avalanches is abundant (Waythomas et al., 2000), and Iliamna has experienced at least 12 very large (horizontal runout length $L > 5$ km) ice–rock avalanches since 1960 (Huggel et al., 2007; Allstadt et al., 2017). 10 of these 12 events occurred on Iliamna's east-facing Red Glacier. These avalanches typically fail in ice or at the ice–bedrock interface near the base of the hydrothermally altered fumarole zones near the summit. The avalanches are relatively frequent, with a recurrence interval of 2–4 years. This interval may be linked to the "recharging" time required for ice thickness to grow until

shear stress exceeds shear strength (Huggel et al., 2007).

Iliamna's ice–rock avalanches have been extensively studied via geologic mapping, multispectral satellite image analysis, numerical modeling, and seismic analysis. Geologic investigations by Waythomas et al. (2000) revealed that late Holocene debris avalanche deposits composed of hydrothermally altered rock are present in most of Iliamna's glacier-filled valleys. From the thin, blanket-like appearance of these deposits, Waythomas et al. (2000) inferred that the original avalanches likely

contained a significant amount of snow or ice in addition to rock. Caplan-Auerbach et al. (2004) documented the seismic signals associated with four very large Iliamna ice–rock avalanches. They found the signals to be remarkably similar, each

Copyright waived. CC0 1.0.

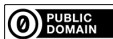



exhibiting a precursory pattern of 20–60 min of repeating discrete events which become closer together in time, culminating with a high-amplitude "spindle" corresponding to the actual failure. This precursory phenomenon was explored further by Caplan-Auerbach and Huggel (2007), who defined four phases of precursory activity:

1. Crevasse opening, with minimal seismic energy release.

2. Acceleration of glacier movement.

3. Discrete slipping, manifested as repeating seismogenic stick-slip events.

4. Continuous slipping, which begins about 0.5–1 h prior to failure.

Caplan-Auerbach and Huggel (2007) also suggested that Iliamna's glaciers are affected by volcanogenic heating, enabling
them to fail on slopes shallower than the 45° threshold broadly assumed to be the minimum slope for cold-ice failure (Huggel et al., 2004). Huggel et al. (2007) found that satellite-derived thermal anomalies in Iliamna's summit region were spatially correlated with zones of fumarolic activity and hydrothermally-altered rocks. Huggel et al. (2007) and Schneider et al. (2010) used successively more sophisticated numerical flow models to reconstruct a very large 2003 Red Glacier avalanche. Both studies were able to recreate flow features persistently observed for Red Glacier events since 1960, such as multiple distal flow
lobes (toes) and prominent superelevation-like flow lobes on the orographically left side of the flow.

## 2.5   The May 2016 and June 2019 ice–rock avalanches

On 22 May 2016 at 07:58 UTC (about midnight local time; hereafter, all times in UTC unless otherwise noted), the Alaska Volcano Observatory (AVO) recorded notable seismic signals on Iliamna's local monitoring network, and a subsequent pilot report confirmed that a large mass movement had occurred. A Landsat 8 image acquired the following day revealed a large
dark-colored deposit on Red Glacier; this deposit also was visible from Homer (Fig. 3a). A horizontal crown-to-toe runout length $L$ of 8.5 km and a vertical drop height $H$ of 1.7 km were estimated from follow-up imagery analysis.

On 21 June 2019 at 00:03 (16:03 on 20 June local time), AVO recorded signals on Iliamna's network indicative of another large avalanche. Photos from citizen overflights taken in the following several days (Fig. 3b and c) showed a large deposit on Red Glacier. Satellite imagery analysis produced values of $L = 8.1$ km and $H = 1.7$ km. The combined source, track, and
deposit areas for these two avalanches are delineated in Fig. 2.

## 3   Data

The 2016 and 2019 Iliamna ice–rock avalanches are well-documented due to the relatively accessible nature of the volcano – by Alaska standards – as well as the exceptional instrument coverage afforded by several permanent and temporary seismoacoustic networks. Our seismoacoustic observations and interpretations were assisted by high-resolution (spatial and temporal) satellite
imagery, aerial and ground-based imagery acquired fortuitously or opportunistically in the days following the events, and high-resolution elevation data.

Copyright waived. CC0 1.0.

Earth **Surface** Dynamics
Discussions
EGU

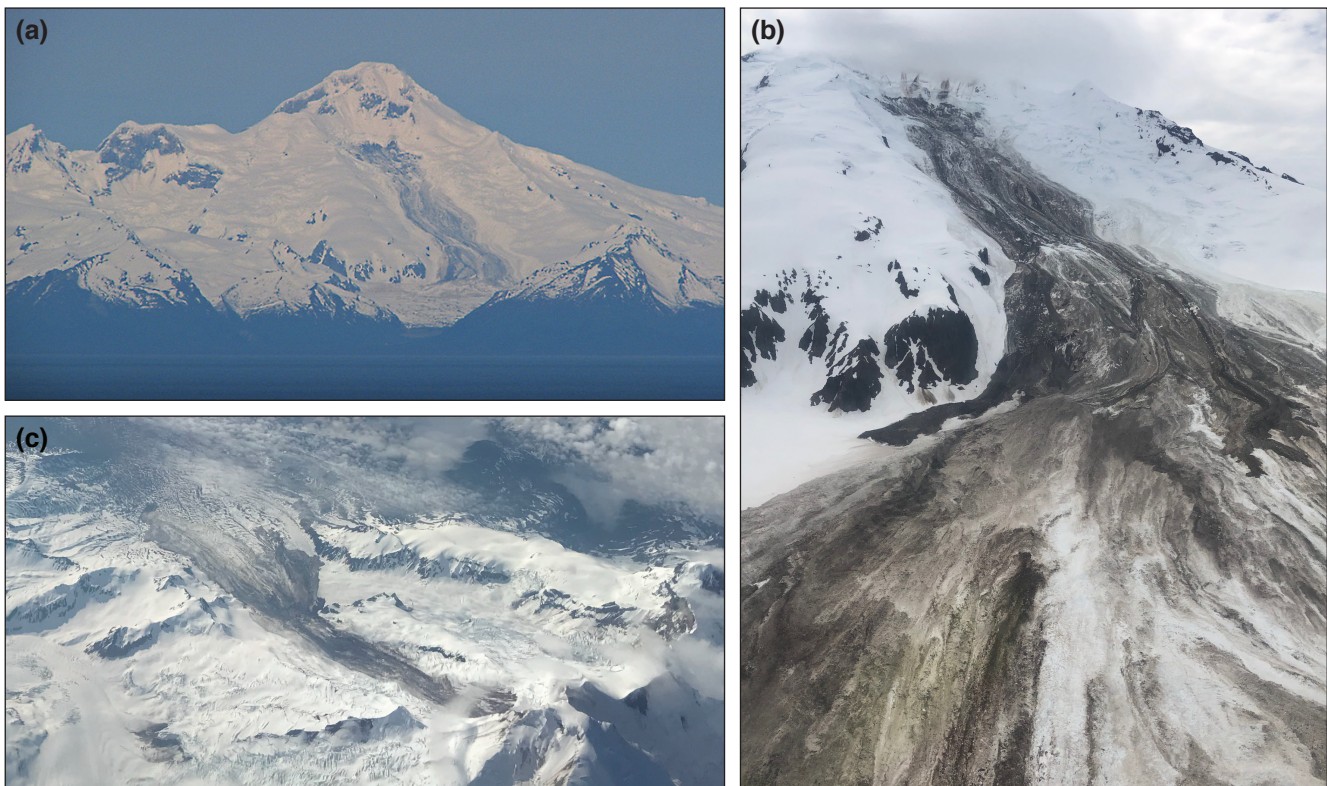

**Figure 3.** Photographs of the 2016 and 2019 Red Glacier avalanche deposits. **(a)** West-northwest-looking photograph of 2016 deposit taken from near Homer on 23 May 2016. Photo courtesy Dennis Anderson, Night Trax Photography; Alaska Volcano Observatory (AVO) image database ID 95521. **(b)** West-northwest-looking aerial photograph of 2019 deposit, 22 June 2019. Photo courtesy Loren Prosser; AVO image database ID 140871. **(c)** Southeast-looking aerial photograph of 2019 deposit, 25 June 2019. Photo courtesy Greg Johnson; AVO image database ID 141431.

### 3.1 Seismic signals

Seismic signals from the events were broadly recorded on local and regional networks. Stations in the EarthScope USArray Transportable Array (network code TA), AVO (network code AV; Power et al., 2020), and Alaska Earthquake Center (AEC; network code AK) networks recorded signals from both events. The temporary Southern Alaska Lithosphere and Mantle Observation Network (SALMON; network code ZE; Tape et al., 2017), which was was deployed from 2015–2017, captured the 2016 event. Additionally, stations in the National Tsunami Warning Center (network code AT), temporary Alaska Amphibious Community Seismic Experiment (network code XO), and Global Seismograph Network (GSN; network code II) networks recorded one or both of the events. Most stations which recorded the signal were broadband (120 s corner period) three-component sensors.

Copyright waived. CC0 1.0.

Earth **Surface**
**Dynamics**
Discussions
EGU
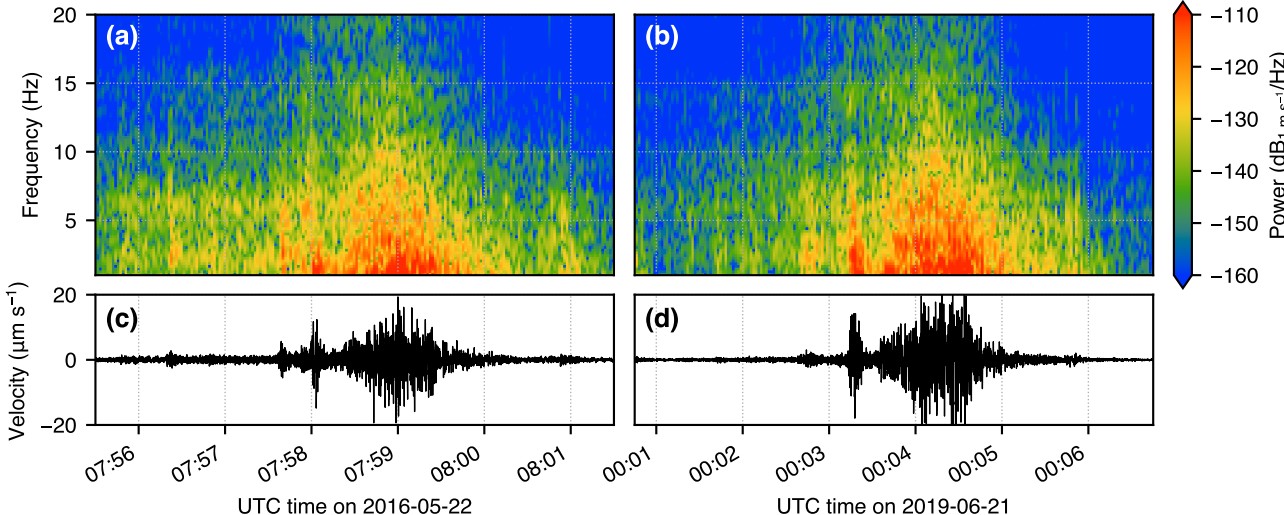

**Figure 4.** Vertical-component spectrograms **(a, b)** and seismic waveforms **(c, d)** from Alaska Volcano Observatory station ILSW for the 2016 (left column) and 2019 (right column) avalanches. Waveforms are highpass filtered at 100 s.

AVO station ILSW, at ∼6 km from the avalanche crowns (Fig. 1), was the closest seismometer with usable data in both 2016 and 2019. We note here that due to the size and mobility of these avalanches, source-to-receiver distances change drastically over the course of the event; ILSW is ∼12 km from the toes of the deposits. Vertical-component spectrograms and waveforms of avalanche seismic signals recorded at this station are shown in Fig. 4. Multiple high-frequency transients are visible in the
spectrograms prior to the main event, indicative of precursory stick-slip activity which has been observed for previous Red Glacier avalanches and is thoroughly explored in Caplan-Auerbach and Huggel (2007). The main event waveforms have an emergent "spindle" shape characteristic of mass movement seismic signals (Allstadt et al., 2018). This same shape, albeit with a lower signal-to-noise ratio (SNR), is found on all stations which recorded the event. The events also produced prodigious long-period energy with a dominant period of 35 s (Fig. 5). In this manuscript we do not analyze the precursory stick-slip
activity observed for the 2016 and 2019 avalanches.

## 3.2 Acoustic signals

The 2016 and 2019 events produced strong infrasound signals which were recorded out to distances exceeding 600 km (Fig. 6). Signals were observed on select infrasound "single station" sensors of the TA, GSN, and AEC networks, as well as regional arrays operated by AVO and one International Monitoring System (network code IM) array. The nearest infrasound sensor at
the time was TA station O20K (Fig. 1) at ∼19 and ∼26 km from the avalanche toes and crowns, respectively.

"Waterfall" plots of the infrasound signal at O20K in different frequency bands for the 2016 and 2019 events are shown in Fig. 7 and illustrate the signal's broadband nature. The dominant frequency of the signal is about 0.5 Hz, but energy exists from

Copyright waived. CC0 1.0.

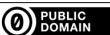



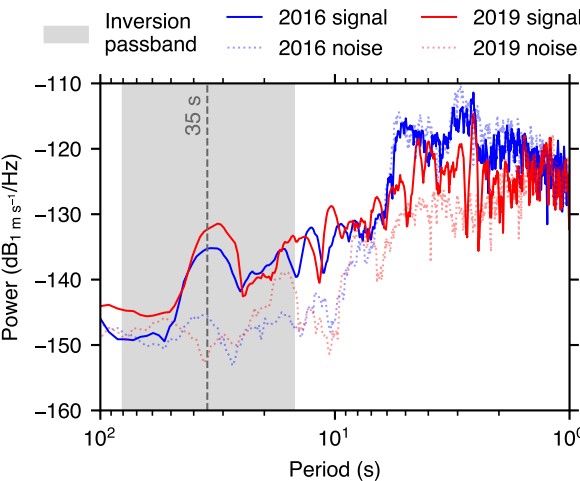

**Figure 5.** Power spectral densities (PSDs) of vertical-component seismic signals from the 2016 (blue lines) and 2019 (red lines) avalanches. The signals were recorded on Alaska Earthquake Center station HOM, the closest station to Iliamna Volcano used in both force inversions. Dotted lines are the PSDs for a 1000 s post-avalanche time window and indicate the approximate contemporaneous noise level. Grey box indicates the force inversion passband (15–80 s).

100 s up to 10 Hz (the Nyquist frequency for this station). In 2016, the ∼120 s duration of the high-frequency signal (2–10 Hz, red line) is nearly twice that of the longer-period signal (0.01–0.1 Hz, blue line). The 2016 and 2019 signals are of similar amplitudes, but in 2019 the noise level is higher in the 0.01–0.1 and 2–10 Hz bands (Fig. 7b).

### 3.3 Aerial photos, satellite imagery, and elevation data

We interpret a wealth of image data to augment our waveform-based analyses. Our satellite image sources are the Planet Labs PlanetScope (3-meter resolution) and RapidEye (5-meter resolution) satellite constellations and the DigitalGlobe WorldView-3 (WV-3, sub-meter resolution) satellite. We use the near-infrared band (NIR) from Planet Labs images each acquired less than 48 h post-event (23 May 2016 and 22 June 2019) to constrain the dimensions of the source and deposit areas for each avalanche (Fig. 2). Fortunately, cloud cover was minimal during this time window. A panchromatic WV-3 image from 22 June 2016 captured the finer details of the source and deposit, though we note that melting of the icy portion of the deposit as well as additional smaller mass movements during the month between the 2016 avalanche and acquisition of the WV-3 image complicate our analysis of the image.

The 2016 and 2019 deposits were readily visible from Homer (Fig. 3a). Members of the community captured oblique aerial photos of the 2019 event during flyovers on 22 and 25 June 2019 (Fig. 3b and c). Additionally, in late July 2019 National Park Service and AVO staff flew a structure from motion (SfM) mission in the area around Iliamna, capturing about 4,400 photos of the edifice and Red Glacier areas that were used to produce a 70 cm resolution digital elevation model (DEM). The DEM

Copyright waived. CC0 1.0.

Earth **Surface** Dynamics Discussions

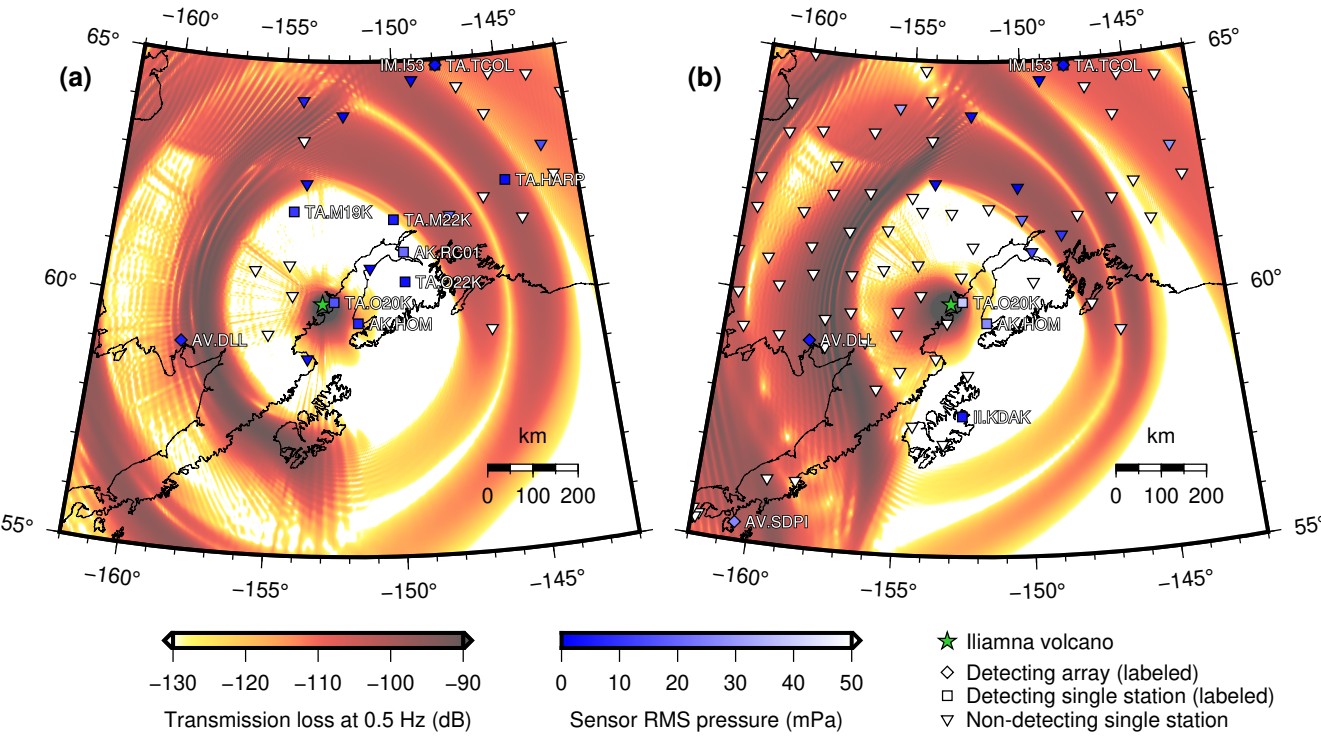

**Figure 6.** Acoustic transmission loss at the Earth's surface, modeled at 0.5 Hz for the **(a)** 2016 and **(b)** 2019 avalanches. The atmospheric model is a single sonde (1D atmospheric profile) over the avalanche path midpoint. Iliamna Volcano is indicated by a green star. Diamonds/squares denote arrays/stations where the avalanche signal was detected. Inverted triangles indicate other infrasound stations where no signal was observed. The blue shades on the station markers indicate root-mean-square (RMS) pressure in the 0.5–2 Hz band for hour-long windows prior to the predicted true arrival. This is a proxy for local site noise. (See §3 for description of network codes.)

extent completely covers the total areas of both events. We use this DEM in our analysis with the caveat that the bed surface
of Red Glacier is highly dynamic due to erosion from mass movements as well as glacial activity; the DEM is therefore more
valid for the 2019 event than the 2016 event.

## 4  Methods

### 4.1  Infrasound analyses

Infrasound signals travel in atmospheric waveguides created primarily by vertical gradients in temperature and horizontal
winds (Drob et al., 2003). The presence or absence of such waveguides in a given propagation direction from the source
strongly controls our ability to detect and characterize infrasonic signals (Fee et al., 2013). Furthermore, cultural and natural
noise, especially locally-sourced wind noise, can obscure a true signal. Just as in seismology, our goal for source studies is

Copyright waived. CC0 1.0.



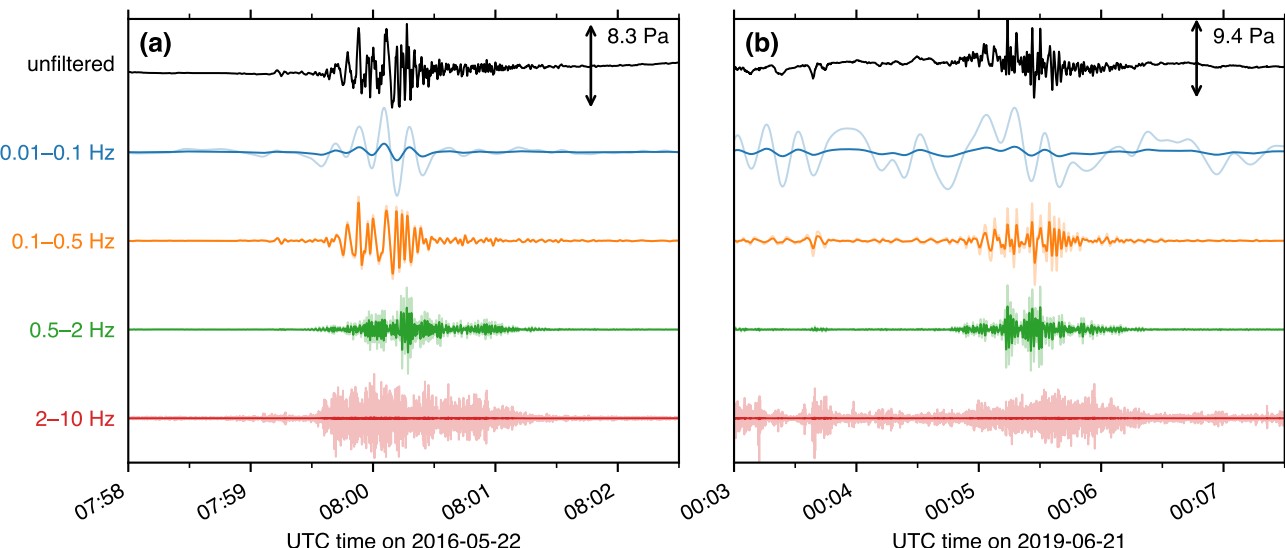

**Figure 7.** Infrasound signals in different frequency bands for the **(a)** 2016 and **(b)** 2019 avalanches. Signals were recorded on Transportable Array station O20K, the closest infrasound sensor to Iliamna Volcano at the time. Signals plotted as solid lines are normalized relative to the black unfiltered trace. Translucent lines are individually normalized signals.

to isolate source properties from path and station effects. To achieve this for the Iliamna avalanches, we model infrasound propagation conditions and assess station noise levels for time periods surrounding each event.

### 4.1.1 Propagation modeling

We use the AVO-G2S (ground-to-space) open-source atmospheric specification (github.com/usgs/volcano-avog2s; Schwaiger et al., 2019) to examine infrasound propagation from the avalanches. We extract a 1D atmospheric profile above the avalanche path midpoint for the forecast hours of 22 May 2016 08:00 and 21 June 2019 00:00. AVO-G2S smoothly merges lower-atmosphere numerical weather prediction (NWP) products with upper-atmosphere empirical climatologies. We use the ERA5 NWP model from the European Centre for Medium-Range Weather Forecasts. The upper atmosphere winds and temperature in AVO-G2S are defined by the 2014 update to the Horizontal Wind Model (Drob et al., 2015) and the NRLMSISE-00 atmospheric model, respectively. The output 1D profile defines temperature, zonal (east–west) and meridional (north–south) winds, density, and pressure as a function of altitude.

We then use the aforementioned profiles and the Modess code from NCPAprop (github.com/chetzer-ncpa/ncpaprop; Waxler et al., 2017) to model infrasonic transmission loss on the Earth's surface in the region around Iliamna. The transmission loss (TL) is the accumulated sound pressure loss as a function of range and height, expressed in decibels (dB). Modess solves a generalized Helmholtz equation for the propagation of a monochromatic pulse in a stratified (i.e., 1D) atmosphere. The method of normal modes is used to solve the equation, which uses the "effective sound speed approximation" – that is, the sum of the

Copyright waived. CC0 1.0.

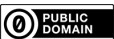



static sound speed and the along-path contribution of the horizontal wind field define the effective sound speed. We choose a
0.5 Hz frequency for modeling, as that is the dominant frequency of the observed acoustic signal, and we set the source height
at 900 m, the approximate elevation of the midpoint of the avalanche paths. We compute the surface acoustic TL in dB from
0–1000 km range for azimuths of 0–360° in 1° increments. We then map the data from range–azimuth space (with the origin
being Iliamna) to longitude–latitude on the WGS84 ellipsoid and grid the result to produce continuous TL maps for the two
events.

### 4.1.2 Noise characterization

To assess the effect of local station noise on signal detection for single infrasound stations, we compute root-mean-square
(RMS) pressure in the 0.5–2 Hz band on hour-long windows for each infrasound-equipped station within 900 km of Iliamna.
We remove the instrument response, detrend, and taper the data prior to filtering. Windows are defined to sample the data in
the hour immediately preceding signal arrival at a given station to avoid possible upwards biasing of extremely quiet stations
by the avalanche signal itself. This is guaranteed by specifying a maximal acoustic celerity (distance / travel time) of 350 m
$s^{-1}$ to define the moveout of the window end time. We remove stations with excessive glitches or dead channels.

### 4.2 Force inversions

We invert the long-period seismic signals generated by these events to obtain the time-varying forces that the avalanche COMs
exerted on the Earth. We use a version of the approach detailed in Allstadt (2013) and applied in Coe et al. (2016). We model the
avalanche as a block sliding down a slope experiencing a net force given by the balance between the slope-parallel gravitational
and dynamic friction forces. By Newton's second law, this net force is equal and opposite to the time-varying force that the
avalanche COM exerts on the Earth (Allstadt, 2013). In our model, this avalanche "force-time function" is applied to the Earth
as a spatially static point force, which is valid for long-wavelength signals where the shift in source location due to mass motion
is small relative to the signal wavelength. We define the point force location to be the COM of the avalanche source region (see
§4.2.4 and yellow stars in Fig. 2).

### 4.2.1 Data selection

We use data from seismic stations within 80–200 km of Iliamna. We omit all stations less than 80 km from the source because
we know from satellite imagery that the COM locations for both events moved up to 8 km. This constraint ensures that we
only use stations for which the source-receiver distance changed by a maximum amount of 10% over the course of the event.
Limiting our station search to 200 km results in a data volume sufficient to constrain the source yet small enough to make
manual signal inspection feasible. Prior to inspection, waveforms were detrended using a second-order polynomial and rotated
into the vertical–radial–transverse (Z–R–T) reference frame. In this frame, radial is defined to be the direction from source
to receiver, and transverse is orthogonal to radial. Both are defined in the horizontal plane. We additionally deconvolve the
instrument response to obtain units of displacement and apply a 15–80 s bandpass filter. The passband was selected to avoid

Copyright waived. CC0 1.0.

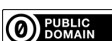



noise associated with the secondary microseism (3–10 s, Gualtieri et al., 2015) and to ensure that the maximum period of the filtered signals is below the corner period of the seismometers used. After this processing, we select waveforms with sufficient SNR by visual inspection. This left us with 28 stations in 2016 and 23 stations in 2019.

### 4.2.2   Predicted ground displacements

The ground displacements at each station are predicted by convolving the force-time function with the Green's functions (GFs)
between the point force location and each station. We use the wavenumber integration method, as implemented in Computer Programs in Seismology (Herrmann, 2013), to calculate the GFs from the ak135 radial Earth velocity model (Kennett et al., 1995). For each station, the GFs describe the 3D displacement as a function of time induced by a parabolic impulse force at the source location. We filter the GFs in the same manner as the data. Mathematically, the three-component ground displacement time series predicted for a station, $\boldsymbol{u}(t) = [u_Z(t),\ u_R(t),\ u_T(t)]$, is given by the convolutions

$$u_Z(t) = [f_N(t)\cos\phi + f_E(t)\sin\phi] * g_{ZH}(t) + f_Z(t) * g_{ZV}(t)\,, \tag{1}$$

$$u_R(t) = [f_N(t)\cos\phi + f_E(t)\sin\phi] * g_{RH}(t) + f_Z(t) * g_{RV}(t)\,,\ \text{and} \tag{2}$$

$$u_T(t) = [f_N(t)\sin\phi - f_E(t)\cos\phi] * g_{TH}(t)\,, \tag{3}$$

where the symbol $*$ denotes convolution (Herrmann, 2013). $\boldsymbol{f}(t) = [f_Z(t),\ f_N(t),\ f_E(t)]$ is the 3D force-time function exerted on the Earth by the avalanche in terms of vertical (Z), north (N), and east (E) components and $\phi$ is the source-to-station azimuth
measured clockwise from north. $g_{ZV}(t)$, $g_{ZH}(t)$, $g_{RV}(t)$, $g_{RH}(t)$, and $g_{TH}(t)$ are the Green's functions describing how vertical (Z), radial (R), and transverse (T) components of displacement are excited by vertical (V) and horizontal (H) force impulses. Note that in the remainder of the text we use $\boldsymbol{f}$ when discussing the force-time function as a model vector – that is, a 1D column vector consisting of the three components of the force-time function concatenated end-to-end. We use $\boldsymbol{f}(t)$ when referring to the 3D force-time function. These two symbols represent the same object. We use the same convention for $\boldsymbol{u}$ and $\boldsymbol{u}(t)$.

### 4.2.3   Mathematical formulation and constraints

In numerical contexts, it is more convenient to formulate the convolution as a matrix multiplication. We therefore transform the GFs into convolution matrices $\boldsymbol{\Lambda}$ by reversing the GFs in time and staggering them as in Allstadt (2013), where the time dependence of the GF is now implicitly stored in the matrix. (For example, the multiplication $\boldsymbol{\Lambda_{ZV}} \boldsymbol{f_Z}$ corresponds to the convolution $f_Z(t) * g_{ZV}(t)$; see Allstadt (2013), Eq. A5.) Making this modification, we can combine Eqs. 1–3 into

$$\boldsymbol{u}^k = \boldsymbol{\Gamma}^k \boldsymbol{f}\,, \tag{4}$$

where now the superscript $k$ denotes the station and $\boldsymbol{\Gamma}^k$ is a matrix of GF convolution matrices:

$$\boldsymbol{\Gamma}^k = \begin{bmatrix} \boldsymbol{\Lambda_{ZV}^k} & \boldsymbol{\Lambda_{ZH}^k}\cos\phi^k & \boldsymbol{\Lambda_{ZH}^k}\sin\phi^k \\ \boldsymbol{\Lambda_{RV}^k} & \boldsymbol{\Lambda_{RH}^k}\cos\phi & \boldsymbol{\Lambda_{RH}^k}\sin\phi \\ \boldsymbol{0} & \boldsymbol{\Lambda_{TH}^k}\sin\phi^k & -\boldsymbol{\Lambda_{TH}^k}\cos\phi^k \end{bmatrix}\,. \tag{5}$$

© Copyright waived. CC0 1.0.

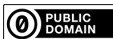



We can now write the linear forward model for $N$ stations as

$$\boldsymbol{d} = \mathbf{G}\boldsymbol{f}, \tag{6}$$

with $\boldsymbol{d} = \begin{bmatrix} \boldsymbol{u}^1, \, \boldsymbol{u}^2, \, \dots, \, \boldsymbol{u}^k, \, \dots, \, \boldsymbol{u}^N \end{bmatrix}^\top$ and $\mathbf{G} = \begin{bmatrix} \boldsymbol{\Gamma}^1, \, \boldsymbol{\Gamma}^2, \, \dots, \, \boldsymbol{\Gamma}^k, \, \dots, \, \boldsymbol{\Gamma}^N \end{bmatrix}^\top$. The superscript $\top$ denotes the transpose; $\boldsymbol{d}$ is a 1D column vector consisting of the data predicted for each component of each station $\boldsymbol{u}^k$ concatenated end-to-end. This is an ill-conditioned problem, so regularization is required to reduce the condition number of $\mathbf{G}$. We invert for $\boldsymbol{f}$ using a higher-order Tikhonov-regularized least squares formulation (e.g., Aster et al., 2013). The solution is

$$\boldsymbol{f} = \left[ \mathbf{G}^\top \mathbf{G} + \alpha^2 \left( a_0 \mathbf{I} + a_1 \mathbf{L}_1{}^\top \mathbf{L}_1 + a_2 \mathbf{L}_2{}^\top \mathbf{L}_2 \right) \right]^{-1} \mathbf{G}^\top \boldsymbol{d}, \tag{7}$$

where $\mathbf{I}$ is the identity matrix and $\mathbf{L}_1$ and $\mathbf{L}_2$ are first- and second-order roughening matrices which approximate the first and second derivatives, respectively. The coefficients $a_0$, $a_1$, and $a_2$ control the degree of importance given to "small," "flat," and "smooth" models, respectively. They must sum to one:

$$\sum_{i=0}^{2} a_i = 1. \tag{8}$$

The regularization parameter $\alpha$ is chosen to balance the constraints on the model specified by the $a_i$ coefficients while still

fitting the data well. We use the L-curve criterion (Hansen, 1992) to find the optimal value for $\alpha$. For both inversions we found the optimal values for these parameters were $\alpha = 4.8 \times 10^{-20}$ and $a_i = [0.4, \, 0, \, 0.6]$. This selection of $a_i$'s prioritizes a model that is both small in magnitude (more centered on zero) and smooth. The inclusion of the higher-order regularization matrices $\mathbf{L}_1$ and $\mathbf{L}_2$ in Eq. 7 separates this method from the method used in Allstadt (2013) and (Coe et al., 2016), which only included zeroth-order Tikhonov regularization.

To characterize the fit of the model to the data, we compute the variance reduction (VR), which is defined as

$$\mathrm{VR} = \left( 1 - \frac{\|\boldsymbol{d} - \boldsymbol{d}_{\mathbf{obs}}\|^2}{\|\boldsymbol{d}_{\mathbf{obs}}\|^2} \right) \times 100\%, \tag{9}$$

where $\boldsymbol{d}_{\mathbf{obs}}$ are the observed data and $\boldsymbol{d}$ are the synthetic data predicted by the forward model (Eq. 6).

In addition to regularization, we constrain all of the components of $\boldsymbol{f}(t)$ to sum to zero to conserve the total momentum of the Earth (see Allstadt, 2013, Appendix A). We also enforce all components of $\boldsymbol{f}(t)$ be zero prior to a specified "zero time."

We choose the zero time to correspond to the point where the vertical component $f_{\mathrm{Z}}(t)$ is non-zero and rising, signaling the initial downward acceleration of the avalanche. The zero time for the 22 May 2016 event is 07:57:53 and the zero time for the 21 June 2019 event is 00:03:08. The selection of the zero time was unambiguous for both events.

To assess the stability of the inversion, we use the jackknife technique (e.g., Moretti et al., 2015; Coe et al., 2016). We run 20 iterations of the inversion, each time randomly discarding 30% of the waveforms.

Copyright waived. CC0 1.0.



### 4.2.4 Trajectory calculations

For simple mass movements, the trajectory can be calculated from the force-time function if the mass is known or can be estimated. The acceleration felt by the avalanche COM is given by Newton's second law

$$\boldsymbol{a}(t) = -\frac{\boldsymbol{f}(t)}{m}, \tag{10}$$

where $m$ and $\boldsymbol{a}(t)$ are the mass and acceleration of the avalanche, respectively. The sign change arises from the fact that $\boldsymbol{f}(t)$ is equal but opposite to the force felt by the avalanche. Integrating twice with respect to time yields the displacement. Since the avalanche paths are straightforward and we have two stable inversions, we apply the double integration method to obtain trajectories for the 2016 and 2019 events. Note that this method assumes that the mass $m$ is constant, which is clearly not the case due to entrainment and deposition along the path. We start integration at the zero time and end at 200 s since the forces are essentially zero at this point. Unlike previous inversions, we add an additional, intuitive constraint that the velocity go to zero at the end of avalanche. This was implemented for each component of the velocity by subtracting a linear trend starting at zero at the zero time and ending at the value of the velocity at 200 s. Note that due to the cumulative effect of double integration, even a small amount of noise occurring early in $\boldsymbol{a}(t)$ can manifest as a large error in the calculated trajectory.

To compare the obtained trajectories with georeferenced data such as satellite imagery and DEMs, we pick a starting location for the COM. Note that the COM start point is not the top of the avalanche crown. We employ a semiautomatic approach in which we use the Planet Labs NIR imagery to estimate the extent of the source region in Google Earth. We define the source region as the zone spreading from the avalanche crown down to where the scoured surface is no longer evident. We then manually outline this region and calculate the centroid of the resulting polygon. Our COM locations are both less than 500 m from the highest point of the avalanche crown; we estimate our error in specifying the COM location to be similar.

We use the satellite imagery shown in Fig. 2 to estimate the mass for each event. First, we subtract the avalanche source area from the total area, ignore entrainment, and assume a uniform 1.5 m deposit thickness everywhere on the slope to obtain a volume. Red Glacier avalanche deposits are typically on the order of a few meters thick (Waythomas et al., 2000; Huggel et al., 2007), so this represents a reasonable estimate. Then we multiply this volume by the density of a mixture of 50% ice (density 920 kg m$^{-3}$) and 50% rock (density 2500 kg m$^{-3}$) to obtain mass estimates. This assumed mixture is based upon the color of the deposits seen in Fig. 2. This approximate calculation yields volumes of 13 million m$^3$ in 2016 and 11 million m$^3$ in 2019. The corresponding masses are 22 billion kg in 2016 and 19 billion kg in 2019.

Two major sources of uncertainty in the trajectory calculations are related to inversion regularization and our mass estimation. The Tikhonov regularization scheme biases the amplitudes of $\boldsymbol{f}$ down from their true values. This means that even if an accurate mass is known, dividing the force-time function by this mass will not recover the true acceleration of the avalanche. We therefore use our mass estimates only to obtain an initial trajectory, which we in turn use to assess reasonable avalanche directionality and shape. To achieve a more realistic trajectory length that is independent of inversion-related biases, we set a target length for the event based on retrospective satellite imagery analysis and iteratively determine the mass that results in this length. The trial mass starts at zero (giving an infinite length) and is increased in increments of 10 million kg until the

Copyright waived. CC0 1.0.

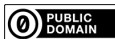



length calculated with the trial mass drops below the target length. The mass obtained via this iterative process is therefore essentially a scaling factor; it is not physically meaningful.

Gualtieri and Ekström (2018) and Schöpa et al. (2018) also performed force inversions using seismic data and inferred masses from deposit imagery. However, in both of these studies the landslides flowed into water, and the authors chose the shoreline as the COM end point. Our COM end points are less clearly defined, since the avalanche mass spread out and formed flow lobes of unknown thickness (Fig. 2). Instead of defining a length by selecting an end point for the COM, which is difficult and subjective due to poor constraints on the thickness of the deposit, we tie salient features in $\boldsymbol{f}(t)$ to consistent features found

in satellite imagery and DEM data, as in Allstadt (2013) and Coe et al. (2016). In particular, we align a prominent northward force in $\boldsymbol{f}(t)$ – which is indicative of the avalanche COM applying such a force to the Earth – with the superelevation-like flow lobe consistently found in both 2016 and 2019 as well as earlier events (see Fig. 2 and Huggel et al., 2007). We then adjust our target length until the location of this northward force aligns with the lobe apex.

## 5   Results

### 5.1   Infrasound

#### 5.1.1   Detection patterns

The 2016 avalanche was detected acoustically on two arrays and eight single stations (Fig. 6a). The 2019 avalanche was detected on three arrays and four single stations (Fig. 6b). We define an array detection as a signal with high correlation (median cross-correlation maxima > 0.6) across the array and a backazimuth pointing towards Iliamna (see e.g. Bishop et al.,

2020, for a discussion of modern array processing techniques). We define a single station detection more qualitatively as a signal with a high SNR in the 0.5–2 Hz band and an acoustic celerity relative to the well-constrained avalanche location and origin time. For both events, at local distances (< 100 km) only stations to the east of Iliamna detected the event. At greater distances (> 200 km) there are detections at many azimuths. The larger number of infrasound stations present in 2019 reflects the westward expansion of the TA deployment, as well as an additional operational AVO array at Sand Point (station code

SDPI, SW corner of Fig. 6b).

#### 5.1.2   Noise characterization

Infrasound station noise levels varied widely (Fig. 6), but all detecting stations in 2016 and 2019 had RMS pressure levels not exceeding 40 mPa in the 0.5–2 Hz band. For both events there are several stations which did not detect the avalanche in spite of having RMS pressure levels less than 40 mPa. Figure 6b reveals that many of the stations installed after the 2016 event were

noisy during the 2019 event, limiting the effective network size increase from 2016 to 2019. For reference, the maximum signal amplitude in the 0.5–2 Hz band at TA station TCOL (the furthest detecting single station from Iliamna) is 18 mPa in 2016 and 23 mPa in 2019.

Copyright waived. CC0 1.0.

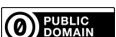



### 5.1.3 Propagation modeling

Figure 6 shows the acoustic TL predicted at the Earth's surface for the 2016 and 2019 events. Dark red bands of lower
TL correspond to ground surface returns from waveguides in the atmosphere, also known as ducts. In general, propagation
conditions differed between the two events within 150 km from Iliamna, becoming more similar at longer ranges. In both
years a strong duct to the west is present, with a low-TL band radially near array DLL. The radial extent of the shadow zone
associated with this duct is similar for both years. However, the local preferred propagation direction differs between 2016 and
2019, with sound being guided to the southeast in 2016 and west-southwest in 2019.

In both years, many stations residing in areas of low predicted TL and therefore higher predicted amplitude (e.g. north of
M19K in 2016 and north and southeast of DLL in 2019) did not detect the event. Conversely, for both events there were stations
which detected the signal despite being located in a predicted shadow zone, such as O22K in 2016 and KDAK in 2019.

## 5.2 Seismic inversion and derivative results

### 5.2.1 Inversion results

The force-time functions for the two events are remarkably similar, showing nearly identical timing and relative amplitude
(Fig. 8a–f). The fits of the modeled data to the true data are displayed in Fig. 9. The variance reduction is 84% for the 2016
inversion and 74% for the 2019 inversion. Grey patches in Fig. 8a–f denote the 95% confidence interval derived from the
jackknife iterations and indicate that our models are not very sensitive to the choice of input waveforms within our dataset. The
overall amplitude of the 2019 event is larger than the 2016 event, consistent with larger seismic waveform amplitudes in 2019
(see Figs. 4c and d and 9). Both results suggest similar durations of about 150 s.

The avalanches initiate with an upward- and westward-directed force, indicating acceleration of the avalanche down and to
the east. This is followed by a complicated yet strikingly similar "coda" for the two events. There are prominent northward
force peaks at ∼40 and ∼80 s. The second is sharper and larger amplitude than the first. There is also a broad southward
force occurring after the first (broad) northward force peak with about the same amplitude, at approximately 65 s. For both
avalanches, the vertical component of $\boldsymbol{f}(t)$ contains two distinct "stair steps" where the force shifts from upwards, to near-zero,
to downwards; these initiate at about 40 and 70 s. Both events conclude with an impulsive downward force occurring at about
100 s in 2016 and 95 s in 2019. After this point, the vertical component is nearly zero, while the horizontal components show
low-amplitude, long-period undulations which are more pronounced in 2019.

### 5.2.2 Trajectories and flow speeds

Seismically-derived avalanche trajectories generally agree with true trajectories for both events. Map and vertical profile views
of the force inversion trajectories for the two events are shown in Fig. 10. As expected given the highly similar force-time
functions, the shapes of the trajectories are very similar. The horizontal displacements indicate that the avalanche COMs
moved almost due east before curving to the south, north, and south again. The vertical profiles in Fig. 10c and d show minor

Copyright waived. CC0 1.0.

Earth **Surface**
**Dynamics**
Discussions

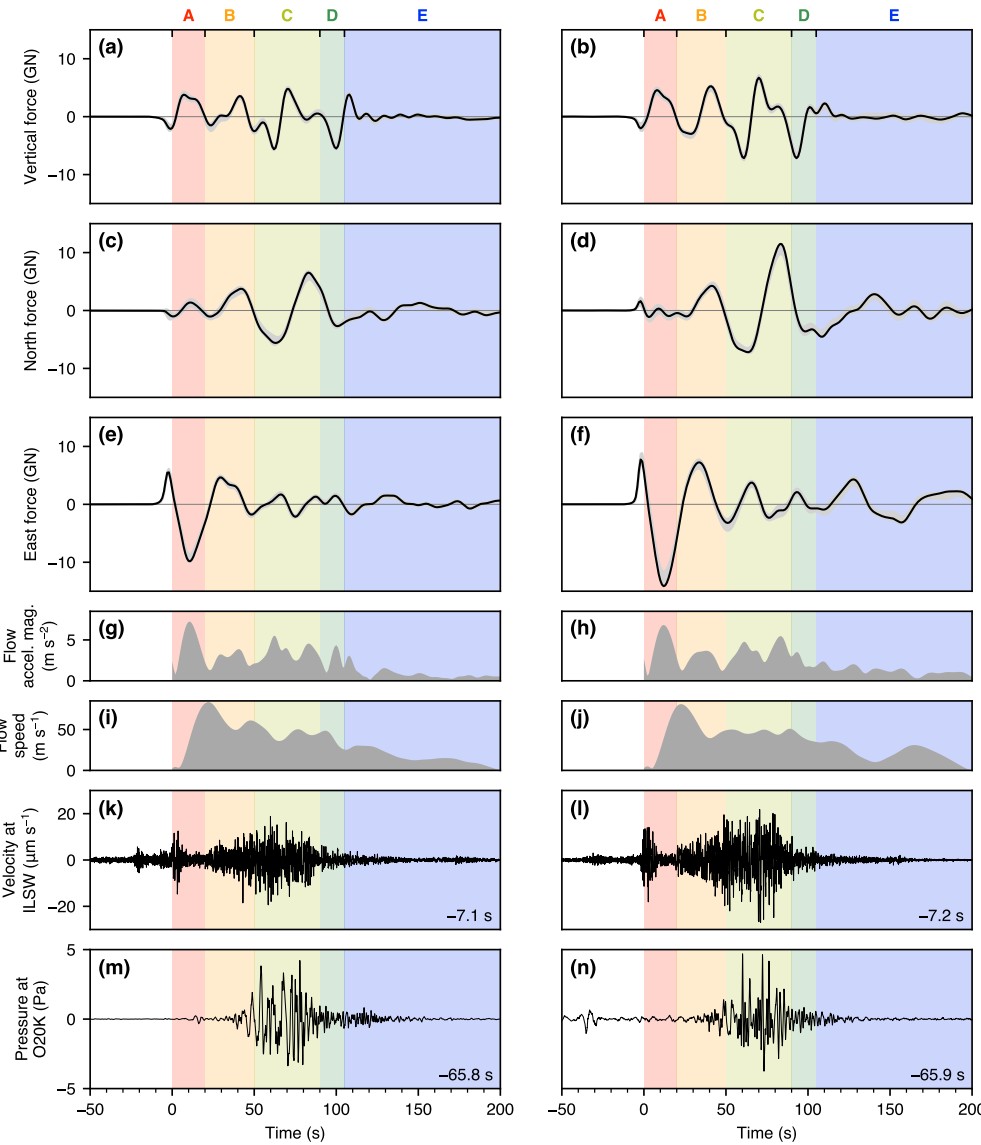

**Figure 8.** Force inversion results for the 2016 (left column) and 2019 (right column) events with seismoacoustic waveforms for reference. **(a–f)** Three-component force-time function $\boldsymbol{f}(t)$. Grey patches show the jackknife-derived 95% confidence interval for $\boldsymbol{f}(t)$. **(g, h)** Force-derived center of mass (COM) acceleration magnitude $\|\boldsymbol{a}(t)\|$. **(i, j)** Force-derived COM speed $\|\boldsymbol{v}(t)\|$. **(k, l)** Vertical component seismic waveforms from station ILSW shifted for travel time from the point force location using a Rayleigh group wavespeed at 1 Hz of 900 m s$^{-1}$. **(m, n)** Infrasound waveforms from station O20K shifted for travel time from the avalanche path midpoint using an acoustic wavespeed at 10 °C of 337 m s$^{-1}$. The time shifts are indicated on the corresponding plots. Seismoacoustic waveforms are highpass filtered at 0.1 Hz. The time axes are relative to the inversion zero time. Colored patches correspond to those in Fig. 10 and letters A–E in §6.2.

Copyright waived. CC0 1.0.

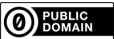



**Figure 9.** Waveform fits for the **(a)** 2016 and **(b)** 2019 force inversions. Observed data are plotted as black lines; modeled data are shown as red lines. Letters in parentheses indicate vertical (Z), radial (R), and transverse (T) components, and distance to the point force location is noted for each waveform. Boldface labels indicate components of stations used in both inversions (see Fig. 1). Waveforms are *not* individually normalized, and the amplitude scale is identical between **(a)** and **(b)**. The time axes are relative to the inversion zero time. (See §3 for description of network codes.)

undulations on an otherwise fairly constant slope, and are strictly decreasing as expected. The black lines are slices through the
SfM DEM along the corresponding horizontal trajectory. The vertical 2016 trajectory (Fig. 10c) and horizontal 2019 trajectory (Fig. 10b) show notable deviations from the DEM and imagery observations – we explore causes for this in §6.5. Jackknifed trajectories, shown as translucent colored lines in Fig. 10, show about 1 km of spread on either side of the true location for the horizontal COM end point. For both events the dominant eastward directionality is evident regardless of jackknife iteration.

Copyright waived. CC0 1.0.

Earth **Surface**
**Dynamics**
Discussions
EGU

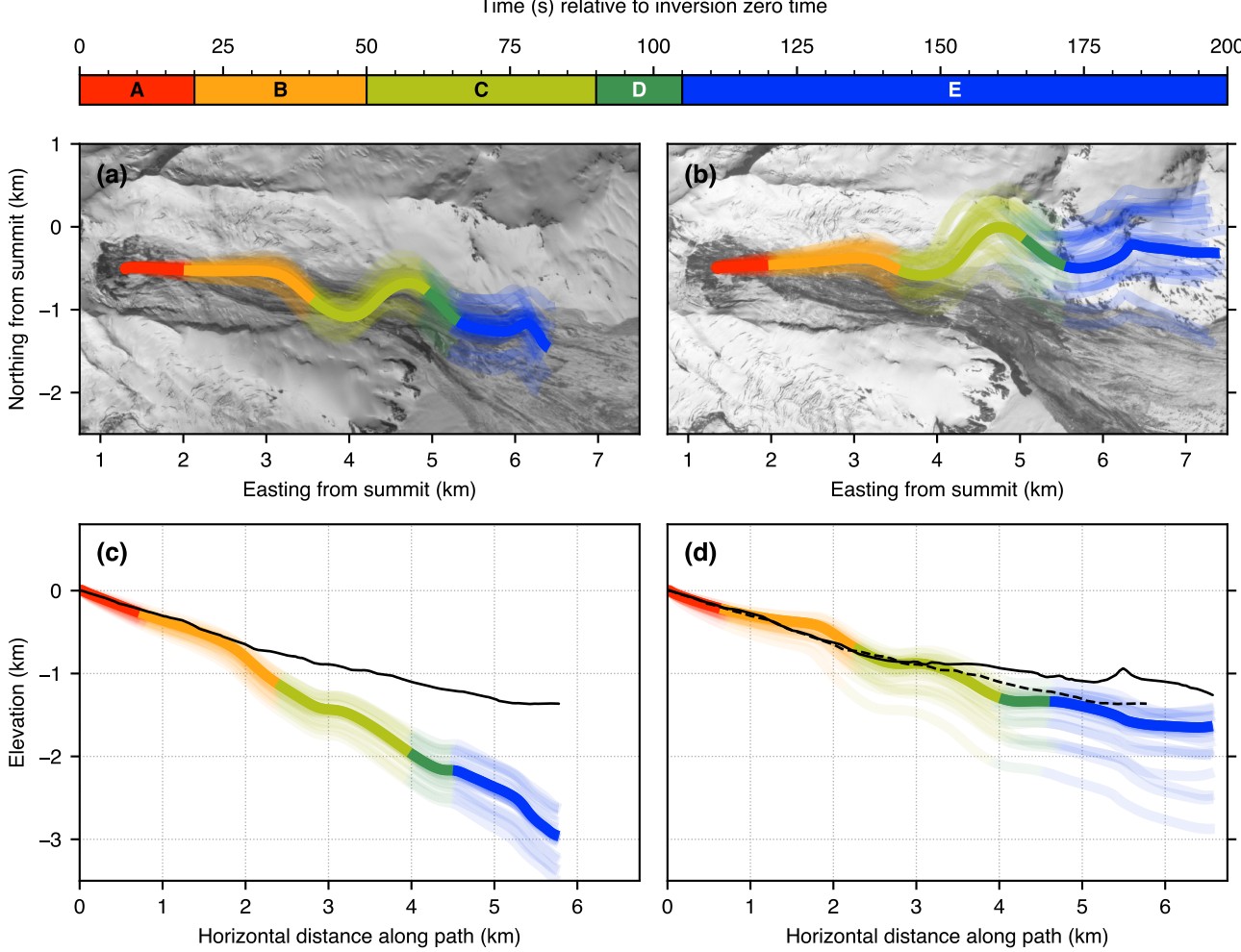

**Figure 10.** Map and profile views of trajectories integrated from the inversion force-time functions for the 2016 **(a, c)** and 2019 **(b, d)** events. Multiple translucent lines correspond to trajectories computed from the jackknife runs. Background images in **(a)** and **(b)** are the same as in Fig. 2. Black lines in **(c)** and **(d)** are profiles through the structure from motion digital elevation model (SfM DEM) along the corresponding horizontal trajectory. Dashed line in **(d)** is the SfM DEM profile from **(c)**. Colored segments correspond to those in Fig. 8 and letters A–E in §6.2. Imagery © 2016 and 2019 Planet Labs, Inc.

Note that the jackknifed trajectories primarily show uncertainties related to station coverage and data selection effects; other sources of trajectory uncertainties which also grow with time are not captured by the jackknife procedure and are discussed in §6.5.

Force-inversion derived COM runout distances and flow speeds have realistic magnitudes and are similar between the two events. Pinning the large northward force $\boldsymbol{f}(t)$ to the flow lobe on the orographically left side of the flow path as described in

Copyright waived. CC0 1.0.

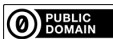



§4.2.4 gives a horizontal along-path COM distance $L_{\text{COM}}$ of 5.8 km with a corresponding mass of 1.4 billion kg for the 2016
event. For the 2019 event, $L_{\text{COM}} = 6.6$ km and the mass is 2.1 billion kg. Both trajectories indicate that most of the avalanche
COM displacement occurred within the first $\sim$150 s of flow (Fig. 10). Average and maximum speeds obtained by integration
of $\boldsymbol{f}(t)$ are 33 and 84 m s$^{-1}$ in 2016 and 34 and 81 m s$^{-1}$ in 2019, respectively.

## 6 Discussion

### 6.1 Acoustic source directionality

We lack sufficient infrasound station coverage to fully test an applicable acoustic source model, such as a single or distributed
dipole, so we do not attempt an acoustic source inversion (e.g., Kim et al., 2012; Iezzi et al., 2019) here. Our infrasound
analysis is therefore largely qualitative. By modeling infrasound propagation and site noise conditions, we sought to isolate
source properties, such as size and directionality, from path effects. For example, Perttu et al. (2020) found that atmospheric
propagation effects could not explain the infrasound radiation pattern observed for the 2018 Anak Krakatau flank collapse, and
used this to infer that the collapse acted like a piston, pushing sound in a directed manner.

For the Iliamna avalanches, examination of acoustic propagation alone might lead one to believe that source directionality is
present, given the consistent detections of stations to the east of Iliamna despite variable local propagation conditions between
the two events. However, there are two complicating factors in our case. Firstly, station noise analysis (Fig. 6) shows that local
stations to the west of Iliamna (and to the north and south as well in 2019) had high noise levels, indicating that preferential
detection on stations to the east could simply be due to lower noise levels at those stations. Secondly, while rugged topography
surrounds Iliamna, there is less topography to the east than to the west (Fig. 1). Furthermore, the avalanches occurred on the
east flank of Iliamna. Since infrasound propagation at local distances is strongly controlled by topography (Kim et al., 2015),
propagation to the east from Iliamna may be topographically preferred. These complicating factors preclude us from assessing
source directionality or obtaining quantitative source estimates.

### 6.2 Multi-stage failure and flow

Synthesis of the force-time function with high-frequency waveforms and force-derived COM acceleration magnitudes and
speeds (Fig. 8) as well as force-derived trajectories (Fig. 10) suggests a consistent multi-stage failure and flow pattern for both
avalanches. Our interpretation is as follows, with approximate times relative to the inversion zero time as well as color codes
given in brackets:

A. Initial failure of the source region in ice or at the ice–bedrock interface and subsequent sliding at an average angle of
$\sim$20°, manifested as a high-frequency seismic transient and a substantial eastward acceleration. No detectable infrasound
is generated by this process (the small pulse visible at $\sim$15 s in 2016 is not seen on any other stations or arrays and is
therefore likely noise) [0–20 s; red].

Copyright waived. CC0 1.0.

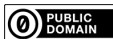

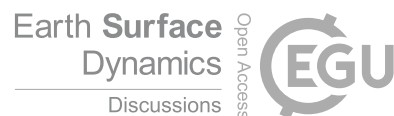

B. The avalanche mass reaches its maximum speed and material becomes fragmented, changing the flow regime from coherent to granular and turbulent. This is manifested as a gradual increase in the high-frequency seismic energy; infrasound energy begins to rise simultaneously as the flow bends to the south [20–50 s; orange].

C. The flow bends to the north and then to the south as both high-frequency seismic and infrasound signals reach their peak amplitudes. Flow speeds decrease but stay between ∼40–60 m s$^{-1}$ [50–90 s; light green].

D. The flow encounters a change to a shallower slope angle (< 10°) where a tributary glacier joins Red Glacier from the southwest (see Fig. 2). This is manifested as an impulsive, relatively short-period (∼30 s) downward force. The high-frequency seismic and infrasound signals taper off and flow speeds continue a slow decline [90–105 s; dark green].

E. After passing the kink in topography, the flow broadens and decelerates, forming the wide, flat debris lobe seen in Fig. 2. The east component of $\boldsymbol{f}(t)$ is largely positive, indicating deceleration of the flow. The vertical component of $\boldsymbol{f}(t)$ is near-zero, and this portion of the flow is not seismically or acoustically energetic for frequencies > 0.1 Hz [105+ s; blue].

Our trajectories are compatible with numerical flow models for Red Glacier performed by Huggel et al. (2007) and Schneider et al. (2010), which both indicate that the avalanche COM tends to the orthographic right and then orthographic left, in the latter case forming a superelevation-like flow lobe visible in Fig. 2. We do not consider the possibility that the observed deposit was formed by two separate flows, as suggested by Huggel et al. (2007) for the 1980 and 2003 Red Glacier avalanches, as we do not see evidence for two separate flows in the seismoacoustic signals or in satellite imagery of the deposits (Fig. 2) and our modeling assuming a single flow is compatible with previous modeling and observations. This suggests that only one flow took place, at least in 2016 and 2019.

### 6.3 Mass estimation

One complication of extracting quantitative information from the force inversion results concerns the method of regularization. Since we impose penalties on the size, slope, and roughness of $\boldsymbol{f}$ via the $a_i$ coefficients, the resultant force amplitudes are likely artificially depressed compared to the true values, as mentioned in §4.2.4. This is evidenced by the much smaller magnitude of the masses from the force inversion trajectories versus our satellite imagery based estimates (1.4 and 2.1 billion kg versus 22 and 19 billion kg for the 2016 and 2019 events, respectively), suggesting that the force amplitudes are indeed being suppressed by the regularization scheme. Additionally, we are inverting a band-limited signal – energy present at very long periods (> 80 s) is not reflected in $\boldsymbol{f}$, which also artificially depresses $\boldsymbol{f}$. Due to these biases, we do not apply the scaling relationship of Ekström and Stark (2013) to these results. We note that in general the masses of these events are not well constrained due to poor constraints on deposit thickness and the relative contributions of entrainment and deposition to the total failure mass. Better groundtruth estimates of avalanche deposit properties would help constrain the effect of regularization, and we encourage such studies in the future. We do note that the phase and *relative* amplitude of the force-time function between the two events (Fig. 8a–f) are not affected by the regularization.

Copyright waived. CC0 1.0.

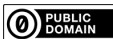


## 6.4 Flow dynamics


Average flow speeds for mass movements can be estimated from the duration of the high-frequency seismic envelope if the horizontal runout length $L$ is known (Caplan-Auerbach et al., 2004). However, it is often difficult to estimate the duration of the flow from the seismic envelope, since the noise floor can bury the earliest and latest parts of the emergent signal (Huggel et al., 2007). Caplan-Auerbach and Huggel (2007) estimated average flow speeds of about 20–50 m s$^{-1}$ for Red Glacier

avalanches using this method. A more complete assessment of flow speeds can be obtained from numerical modeling or by examination of the speed time series $\|\boldsymbol{v}(t)\|$ obtained from the force inversion, since these provide both average and maximum values. Schneider et al. (2010) found an average flow speed of about 50 m s$^{-1}$ and peak flow speeds between 70–100 m s$^{-1}$ for a numerically modeled 2003 Red Glacier avalanche. Our values derived from $\|\boldsymbol{v}(t)\|$ (average speeds of 33 and 34 m s$^{-1}$ and maximum speeds of 84 and 81 m s$^{-1}$ in 2016 and 2019, respectively) are compatible with these results as well

as those of Caplan-Auerbach and Huggel (2007), though we note that our values describe COM dynamics, while those of Caplan-Auerbach and Huggel (2007) are calculated using $L$ and therefore apply to the flow front. Our results are also broadly compatible with other studies of similar large avalanches, such as the July 2007 Mount Steele rock–ice avalanche (35–65 m s$^{-1}$ average speed; Lipovsky et al., 2008) and the June 2016 Lamplugh Glacier rock avalanche ($\sim$55 m s$^{-1}$ maximum speed; Dufresne et al., 2019).

Inversion-derived COM acceleration magnitudes $\|\boldsymbol{a}(t)\|$ and flow speeds $\|\boldsymbol{v}(t)\|$ are plotted in Fig. 8g–j. The peak infrasound amplitude does not correlate with peak acceleration magnitude nor peak speed, instead occurring about 50 s after the latter. This notable latency between peak speed and peak acoustic energy might be explained by a model similar to Marchetti et al. (2019b), where infrasound is produced by waves at the free surface of the flow. Such waves would take time to develop since the initially blocky mass needs to be sufficiently fragmented and turbulent, which requires high flow speeds.

Another possibility is that flow interaction with a particular topographic feature along the flow path is generating infrasound, and that the observed peak amplitude timing corresponds to the travel time for the flow to reach this feature. Figure 8m and n indicates that peak infrasound occurs anywhere from 60–85 s into the flow. Since the prominent northward force linked to the flow lobe on the orographically left side of the flow occurs at about 80 s, flow turbulence at this point could be responsible for the peak in infrasound energy. Moore et al. (2017) observed a ground-coupled air wave associated with the second of two very

large rock avalanches at Bingham Canyon Mine (Utah, USA). They inferred from the timing of the phase that the air wave was likely coupled into the ground when the rock avalanche was beginning to impact the pit bottom, $\sim$50 s after the start of the event. However, this explanation makes less sense in the context of Red Glacier avalanches since the topography of Red Glacier is far smoother (e.g., compare the black line in Fig. 10c to Moore et al. (2017), Fig. 5).

## 6.5 Inversion stability and trajectory uncertainties

The low variance of the jackknife iterations (Fig. 8a–f) indicates that the inversion result is largely unaffected by changes to the input data. We note two prominent issues with the calculated trajectories, however:

Copyright waived. CC0 1.0.



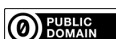

1. The 2019 horizontal trajectory is rotated approximately 15° counter-clockwise relative to the 2016 trajectory, though both have the same shape.

2. The 2016 vertical trajectory is too steep relative to the bed topography (black line in Fig. 10a).

There are several potential causes for these discrepancies. One possibility is that our model loses validity over the course of the event. Since the premise of the force inversion assumes a single point force, as the avalanche moves downslope and transitions from a sliding block to a more distributed, fragmented flow our source model becomes less applicable (see Coe et al., 2016). The vertical trajectories for both 2016 and 2019 show good agreement with the topography for the first ∼50 s of flow, before deviating significantly. This portion of the force history corresponds to the initial sliding block phase, where our

model is most valid, supporting the notion that our model validity decreases with time. However, the horizontal trajectories provide a reasonable quantitative estimate for the entire flow path, not just the initial period of supposed higher model validity. Ultimately, without video footage of the events, improved mass estimates, or sophisticated flow modeling, understanding where the model may begin to break down is challenging.

Another factor is noise within the passband of our inversion. We note that while the SNR for the longer-period portion of

the inversion passband was generally greater in 2019 than in 2016, the SNR for shorter periods (15–25 s) was lower in 2019 than in 2016 (Fig. 5). This greater short-period noise in 2019 is visible when comparing the waveforms in 2016 (Fig. 9a) to those in 2019 (Fig. 9b). We were unable to avoid this noise without increasing the minimum period of the inversion and thereby sacrificing short-period details in $\boldsymbol{f}(t)$, which are consistent between the two events and thus not spurious. Since this short-period noise is more prominent in 2019 than 2016, it could contribute to the misaligned horizontal trajectory for the 2019

event. We note that the VR for the 2019 inversion is about 10% lower than the VR for the 2016 inversion; this is readily seen in Fig. 9.

Finally, our inversion may be biased by uneven azimuthal station coverage or an uneven distribution of seismometer components. For most stations, horizontal components tended to be noisier than vertical components. Consequently, most of our input waveform data for the inversion is vertical component (see component labels in Fig. 9). Mathematically, Eq. 1 shows that given

sufficient azimuthal coverage $\boldsymbol{f}(t)$ should be recoverable from the vertical displacement time series $u_{\mathrm{Z}}(t)$ alone. However, our largely vertical-component input data could be biasing our $f_{\mathrm{Z}}(t)$ amplitudes too high. This in turn would produce overly steep vertical trajectories. We tested the inversion's sensitivity to azimuthal station coverage and found that the 2019 trajectory showed negligible change unless significant deviations (e.g., only retaining stations to the south of Iliamna) were undertaken.

All of the preceding issues are exacerbated when we doubly integrate $\boldsymbol{f}(t)$ to displacement. Therefore, a relatively small

southward bias in $\boldsymbol{f}(t)$ could nudge the entire trajectory northward in the manner seen for the 2019 event. This also applies to the overly steep vertical trajectory in 2016 – if at any point in $\boldsymbol{f}(t)$ the vertical component is overestimated, the vertical trajectory will be affected from that point onwards. In spite of these issues, the consistent shape of the trajectories and the strikingly similar phase and relative amplitude of the force-time functions give us confidence in our modeling.

A key benefit of modeling two highly similar avalanches is the opportunity to compare the inversion results, determine which

features are consistent between the two years, and evaluate the inversion technique. Examination of $\boldsymbol{f}(t)$ for the 2016 event

Copyright waived. CC0 1.0.

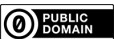



alone might lead one to conclude that the shorter-period details are just spurious byproducts of noise or path effects. However, the 2019 avalanche has flow and deposit characteristics that are remarkably similar to those of the 2016 event, and we observe similar details in $\boldsymbol{f}(t)$ in spite of varying path effects due to different station configurations in 2016 and 2019. This provides confidence in the inversion method used here.

## 7 Conclusions

Surficial mass movements transfer energy into the solid Earth and the atmosphere, producing seismoacoustic signals that yield complementary information about event dynamics. In this study, we took advantage of an extraordinary seismoacoustic dataset from two large, highly similar ice–rock avalanches to reconstruct the dynamics of the events. The repeatability of these avalanches provides an excellent opportunity to test the robustness of our modeling methods. Our force-time functions are derived from the inversion of long-period (15–80 s) seismic signals recorded on stations > 80 km from the avalanches. They indicate that over the course of about 150 s the avalanche COM slid to the east, was subsequently deflected slightly to the south and then to the north, and then broadly decelerated. Our results provide constraints on time-varying avalanche acceleration, velocity, and directionality. This is important for hazard mitigation as well as general understanding of seismic signals from mass movements, though better estimates of mass and flow properties from field studies (e.g., Dufresne et al., 2019) and numerical modeling (e.g., Moretti et al., 2012) are needed to fully exploit this method's potential.

While it was possible to model the avalanche seismic source, we lacked sufficient infrasound data to quantitatively characterize the acoustic source. After accounting for propagation effects and station noise, we cannot assess whether the Iliamna avalanches exhibit acoustic source directionality. Still, the acoustic data are qualitatively consistent with our force-derived reconstructions. It appears that infrasound from these avalanches is produced after the mass movement regime transitions from cohesive block-type failure to granular and turbulent flow, but controlled experiments and denser acoustic instrumentation are needed to test this hypothesis.

Iliamna Volcano is an excellent natural laboratory for the seismoacoustic and geomorphological study of these impressive avalanches due to their relatively frequent occurrence at the volcano. Future work at Iliamna – as well as at other sites of repetitive surficial mass movements – should synthesize advanced numerical modeling techniques with detailed groundtruth information including video footage and repeat high-resolution DEM acquisitions. These efforts, combined with more complete acoustic station coverage – perhaps with arrays as well as single sensors – could result in a substantial increase in our understanding of large debris avalanches and other mass movements. This insight may then be applicable for mitigation of, and response to, the significant hazards posed by these catastrophic surface processes.

*Data availability.* All of the seismic and infrasound data used in this study are available from the Incorporated Research Institutions for Seismology Data Management Center (IRIS DMC).

© Copyright waived. CC0 1.0.

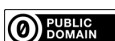



*Author contributions.* L. Toney performed the analysis and wrote the manuscript. D. Fee conceived the idea and advised the project. K. Allstadt wrote the force inversion code. M. Haney supplied input on the inversion process. R. Matoza provided guidance on infrasound analysis. All authors contributed to the direction of the study and the refinement of the manuscript.

*Competing interests.* The authors declare no competing interests.

*Disclaimer.* This draft manuscript is distributed solely for purposes of scientific peer review. Its content is deliberative and predecisional. Because the manuscript has not yet been approved for publication by the U.S. Geological Survey (USGS), it does not represent any official USGS finding or policy. Any use of trade, firm, or product names is for descriptive purposes only and does not imply endorsement by the U.S. Government.

*Acknowledgements.* We used the Python seismological framework ObsPy (www.obspy.org; Beyreuther et al., 2010) extensively in this
study. We also used the spectral estimation Python wrapper mtspec (krischer.github.io/mtspec; Krischer, 2016) to generate the PSDs in Fig. 5. Figures were created using Matplotlib (matplotlib.org; Hunter, 2007), the Generic Mapping Tools (GMT; www.generic-mapping-tools.org; Wessel et al., 2019), and PyGMT (www.pygmt.org; Uieda and Wessel, 2019), the Python interface for GMT.

We thank Hannah Dietterich and Tim Orr for their contributions to the Iliamna SfM project. Jacqueline Caplan-Auerbach and Chris Waythomas helped us understand the recent history and typical character of Iliamna avalanches. Taryn Lopez provided us with information
about zones of recent fumarolic activity on Iliamna. We thank Carl Tape for early access to embargoed SALMON data.

R. Matoza was supported by NSF grants EAR-1614855 and EAR-1847736. The authors also acknowledge support from NSF grant EAR-1614323 and from AVO through the U.S. Geological Survey Volcano Hazards Program.

Ch'naqal'in (Iliamna Volcano) is located on the traditional *etnana* (land) of the Dena'ina people of south-central Alaska.

© Copyright waived. CC0 1.0.

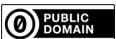



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

Copyright waived. CC0 1.0.

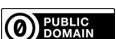



Fee, D. and Matoza, R. S.: An overview of volcano infrasound: From hawaiian to plinian, local to global, J. Volcanol. Geoth. Res., 249, 123–139, https://doi.org/10.1016/j.jvolgeores.2012.09.002, http://dx.doi.org/10.1016/j.jvolgeores.2012.09.002http://linkinghub.elsevier.com/retrieve/pii/S0377027312002685, 2013.

Fee, D., Waxler, R., Assink, J., Gitterman, Y., Given, J., Coyne, J., Mialle, P., Garces, M., Drob, D., Kleinert, D., Hofstetter, R., and Grenard, P.: Overview of the 2009 and 2011 Sayarim Infrasound Calibration Experiments, J. Geophys. Res. Atmos., 118, 6122–6143, https://doi.org/10.1002/jgrd.50398, http://doi.wiley.com/10.1002/jgrd.50398, 2013.

Gualtieri, L. and Ekström, G.: Broad-band seismic analysis and modeling of the 2015 Taan Fjord, Alaska landslide using Instaseis, Geophys. J. Int., 213, 1912–1923, https://doi.org/10.1093/gji/ggy086, https://academic.oup.com/gji/article/213/3/1912/4923053, 2018.

Gualtieri, L., Stutzmann, E., Capdeville, Y., Farra, V., Mangeney, A., and Morelli, A.: On the shaping factors of the secondary microseismic wavefield, J. Geophys. Res. Solid Earth, 120, 6241–6262, https://doi.org/10.1002/2015JB012157, https://onlinelibrary.wiley.com/doi/abs/10.1002/2015JB012157, 2015.

Haney, M. M., Matoza, R. S., Fee, D., and Aldridge, D. F.: Seismic equivalents of volcanic jet scaling laws and multipoles in acoustics, Geophys. J. Int., 213, 623–636, https://doi.org/10.1093/gji/ggx554, https://academic.oup.com/gji/article/213/1/623/4772873, 2018.

Hansen, P. C.: Analysis of Discrete Ill-Posed Problems by Means of the L-Curve, SIAM Rev., 34, 561–580, https://doi.org/10.1137/1034115, http://epubs.siam.org/doi/10.1137/1034115, 1992.

Havens, S., Marshall, H.-P., Johnson, J. B., and Nicholson, B.: Calculating the velocity of a fast-moving snow avalanche using an infrasound array, Geophys. Res. Lett., 41, 6191–6198, https://doi.org/10.1002/2014GL061254, http://doi.wiley.com/10.1002/2014GL061254, 2014.

Herrick, J. A., Neal, C. A., Cameron, C. E., Dixon, J. P., and McGimsey, R. G.: 2012 Volcanic activity in Alaska–Summary of events and response of the Alaska Volcano Observatory, U.S. Geological Survey Scientific Investigations Report 2014-5160, 81 p., https://doi.org/http://dx.doi.org/10.3133/sir20145160, 2014.

Herrmann, R. B.: Computer Programs in Seismology: An Evolving Tool for Instruction and Research, Seismol. Res. Lett., 84, 1081–1088, https://doi.org/10.1785/0220110096, https://pubs.geoscienceworld.org/srl/article/84/6/1081-1088/315307, 2013.

Hibert, C., Ekström, G., and Stark, C. P.: The relationship between bulk-mass momentum and short-period seismic radiation in catastrophic landslides, J. Geophys. Res. Earth Surf., 122, 1201–1215, https://doi.org/10.1002/2016JF004027, http://doi.wiley.com/10.1002/2016JF004027, 2017.

Huggel, C., Haeberli, W., Kääb, A., Bieri, D., and Richardson, S.: An assessment procedure for glacial hazards in the Swiss Alps, Can. Geotech. J., 41, 1068–1083, https://doi.org/10.1139/t04-053, http://www.nrcresearchpress.com/doi/10.1139/t04-053, 2004.

Huggel, C., Caplan-Auerbach, J., Waythomas, C. F., and Wessels, R. L.: Monitoring and modeling ice-rock avalanches from ice-capped volcanoes: A case study of frequent large avalanches on Iliamna Volcano, Alaska, J. Volcanol. Geoth. Res., 168, 114–136, https://doi.org/10.1016/j.jvolgeores.2007.08.009, 2007.

Hungr, O., Leroueil, S., and Picarelli, L.: The Varnes classification of landslide types, an update, Landslides, 11, 167–194, https://doi.org/10.1007/s10346-013-0436-y, http://link.springer.com/10.1007/s10346-013-0436-y, 2014.

Hunter, J. D.: Matplotlib: A 2D Graphics Environment, Comput. Sci. Eng., 9, 90–95, https://doi.org/10.1109/MCSE.2007.55, http://ieeexplore.ieee.org/document/4160265/, 2007.

Iezzi, A. M., Fee, D., Kim, K., Jolly, A. D., and Matoza, R. S.: Three-Dimensional Acoustic Multipole Waveform Inversion at Yasur Volcano, Vanuatu, J. Geophys. Res. Solid Earth, 124, 8679–8703, https://doi.org/10.1029/2018JB017073, https://onlinelibrary.wiley.com/doi/abs/10.1029/2018JB017073, 2019.



Johnson, J. B. and Palma, J. L.: Lahar infrasound associated with Volcán Villarrica's 3 March 2015 eruption, Geophys. Res. Lett., 42, 6324–6331, https://doi.org/10.1002/2015GL065024, http://doi.wiley.com/10.1002/2015GL065024, 2015.

Johnson, J. B. and Ronan, T. J.: Infrasound from volcanic rockfalls, J. Geophys. Res. Solid Earth, 120, 8223–8239, https://doi.org/10.1002/2015JB012436, https://onlinelibrary.wiley.com/doi/abs/10.1002/2015JB012436, 2015.

Kawakatsu, H.: Centroid single force inversion of seismic waves generated by landslides, J. Geophys. Res., 94, 12 363, https://doi.org/10.1029/JB094iB09p12363, http://doi.wiley.com/10.1029/JB094iB09p12363, 1989.

Kennett, B. L. N., Engdahl, E. R., and Buland, R.: Constraints on seismic velocities in the Earth from traveltimes, Geophys.

J. Int., 122, 108–124, https://doi.org/10.1111/j.1365-246X.1995.tb03540.x, https://academic.oup.com/gji/article-lookup/doi/10.1111/j.1365-246X.1995.tb03540.x, 1995.

Kim, K., Lees, J. M., and Ruiz, M.: Acoustic multipole source model for volcanic explosions and inversion for source parameters, Geophys. J. Int., 191, 1192–1204, https://doi.org/10.1111/j.1365-246X.2012.05696.x, https://academic.oup.com/gji/article/191/3/1192/559729, 2012.

Kim, K., Fee, D., Yokoo, A., and Lees, J. M.: Acoustic source inversion to estimate volume flux from volcanic explosions, Geophys. Res.

Lett., 42, 5243–5249, https://doi.org/10.1002/2015GL064466, http://doi.wiley.com/10.1002/2015GL064466, 2015.

Kogelnig, A., Hübl, J., Suriñach, E., Vilajosana, I., and McArdell, B. W.: Infrasound produced by debris flow: propagation and frequency content evolution, Nat. Hazards, 70, 1713–1733, https://doi.org/10.1007/s11069-011-9741-8, http://link.springer.com/10.1007/s11069-011-9741-8, 2014.

Krischer, L.: mtspec Python wrappers 0.3.2, https://doi.org/10.5281/zenodo.321789, 2016.

Lipovsky, P. S., Evans, S. G., Clague, J. J., Hopkinson, C., Couture, R., Bobrowsky, P., Ekström, G., Demuth, M. N., Delaney, K. B., Roberts, N. J., Clarke, G., and Schaeffer, A.: The July 2007 rock and ice avalanches at Mount Steele, St. Elias Mountains, Yukon, Canada, Landslides, 5, 445–455, https://doi.org/10.1007/s10346-008-0133-4, http://link.springer.com/10.1007/s10346-008-0133-4, 2008.

Marchetti, E., Ripepe, M., Ulivieri, G., and Kogelnig, A.: Infrasound array criteria for automatic detection and front velocity estimation of snow avalanches: towards a real-time early-warning system, Nat. Hazard Earth Sys., 15, 2545–2555, https://doi.org/10.5194/nhess-15-2545-2015, https://www.nat-hazards-earth-syst-sci.net/15/2545/2015/, 2015.

Marchetti, E., van Herwijnen, A., Christen, M., Silengo, M. C., and Barfucci, G.: Seismo-acoustic energy partitioning of a powder snow avalanche, Earth Surf. Dynam. Discuss., Discussion Paper, 1–18, https://doi.org/10.5194/esurf-2019-61, 2019a.

Marchetti, E., Walter, F., Barfucci, G., Genco, R., Wenner, M., Ripepe, M., McArdell, B., and Price, C.: Infrasound Array Analysis of Debris Flow Activity and Implication for Early Warning, J. Geophys. Res. Earth Surf., 124, 567–587, https://doi.org/10.1029/2018JF004785,

https://onlinelibrary.wiley.com/doi/abs/10.1029/2018JF004785, 2019b.

Matoza, R., Fee, D., Green, D., and Mialle, P.: Volcano Infrasound and the International Monitoring System, in: Infrasound Monit. Atmos. Stud., edited by Le Pichon, A., Blanc, E., and Hauchecorne, A., pp. 1023–1077, Springer International Publishing, Cham, https://doi.org/10.1007/978-3-319-75140-5_33, https://doi.org/10.1007/978-3-319-75140-5{_}33http://link.springer.com/10.1007/978-3-319-75140-5{_}33, 2019.

Moore, J. R., Pankow, K. L., Ford, S. R., Koper, K. D., Hale, J. M., Aaron, J., and Larsen, C. F.: Dynamics of the Bingham Canyon rock avalanches (Utah, USA) resolved from topographic, seismic, and infrasound data, J. Geophys. Res. Earth Surf., 122, 615–640, https://doi.org/10.1002/2016JF004036, https://onlinelibrary.wiley.com/doi/abs/10.1002/2016JF004036, 2017.

Moran, S. C., Matoza, R. S., Garcés, M. A., Hedlin, M. A. H., Bowers, D., Scott, W. E., Sherrod, D. R., and Vallance, J. W.: Seismic and acoustic recordings of an unusually large rockfall at Mount St. Helens, Washington, Geophys. Res. Lett., 35, L19 302,

https://doi.org/10.1029/2008GL035176, http://doi.wiley.com/10.1029/2008GL035176, 2008.

Copyright waived. CC0 1.0.

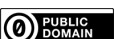



Moretti, L., Mangeney, A., Capdeville, Y., Stutzmann, E., Huggel, C., Schneider, D., and Bouchut, F.: Numerical modeling of the Mount Steller landslide flow history and of the generated long period seismic waves, Geophys. Res. Lett., 39, n/a–n/a, https://doi.org/10.1029/2012GL052511, http://doi.wiley.com/10.1029/2012GL052511, 2012.

Moretti, L., Allstadt, K., Mangeney, A., Capdeville, Y., Stutzmann, E., and Bouchut, F.: Numerical modeling of the Mount Meager landslide constrained by its force history derived from seismic data, J. Geophys. Res. Solid Earth, 120, 2579–2599, https://doi.org/10.1002/2014JB011426, http://doi.wiley.com/10.1002/2014JB011426, 2015.

Perttu, A., Caudron, C., Assink, J., Metz, D., Tailpied, D., Perttu, B., Hibert, C., Nurfiani, D., Pilger, C., Muzli, M., Fee, D., Andersen, O., and Taisne, B.: Reconstruction of the 2018 tsunamigenic flank collapse and eruptive activity at Anak Krakatau based on eyewitness reports, seismo-acoustic and satellite observations, Earth Planet. Sc. Lett., 541, 116 268, https://doi.org/10.1016/j.epsl.2020.116268, https://doi.org/10.1016/j.epsl.2020.116268https://linkinghub.elsevier.com/retrieve/pii/S0012821X20302119, 2020.

Plafker, G. and Ericksen, G. E.: Nevados Huascarán avalanches, Peru, in: Dev. Geotech. Eng., edited by Voight, B., vol. 14, chap. 8, pp. 277–314, Elsevier, https://doi.org/10.1016/B978-0-444-41507-3.50016-7, https://linkinghub.elsevier.com/retrieve/pii/B9780444415073500167, 1978.

Poppeliers, C., Wheeler, L. B., and Preston, L.: The Effects of Atmospheric Models on the Estimation of Infrasonic Source Functions at the Source Physics Experiment, B. Seismol. Soc. Am., pp. 1–13, https://doi.org/10.1785/0120190241, https://pubs.geoscienceworld.org/ssa/bssa/article/583440/The-Effects-of-Atmospheric-Models-on-the, 2020.

Power, J. A., Haney, M. M., Botnick, S. M., Dixon, J. P., Fee, D., Kaufman, A. M., Ketner, D. M., Lyons, J. J., Parker, T., Paskievitch, J. F., Read, C. W., Searcy, C., Stihler, S. D., Tepp, G., and Wech, A. G.: Goals and Development of the Alaska Volcano Observatory Seismic Network and Application to Forecasting and Detecting Volcanic Eruptions, Seismol. Res. Lett., https://doi.org/10.1785/0220190216, https://pubs.geoscienceworld.org/ssa/srl/article/579924/Goals-and-Development-of-the-Alaska-Volcano, 2020.

Ripepe, M., De Angelis, S., Lacanna, G., Poggi, P., Williams, C., Marchetti, E., Donne, D. D., and Ulivieri, G.: Tracking Pyroclastic Flows at Soufrière Hills Volcano, Eos, Trans. Am. Geophys. Union, 90, 229, https://doi.org/10.1029/2009EO270001, http://doi.wiley.com/10.1029/2009EO270001, 2009.

Ripepe, M., De Angelis, S., Lacanna, G., and Voight, B.: Observation of infrasonic and gravity waves at Soufrière Hills Volcano, Montserrat, Geophys. Res. Lett., 37, 1–5, https://doi.org/10.1029/2010GL042557, http://doi.wiley.com/10.1029/2010GL042557, 2010.

Roman, D. C., Power, J. A., Moran, S. C., Cashman, K. V., Doukas, M. P., Neal, C. A., and Gerlach, T. M.: Evidence for dike emplacement beneath Iliamna Volcano, Alaska in 1996, J. Volcanol. Geoth. Res., 130, 265–284, https://doi.org/10.1016/S0377-0273(03)00302-0, https://linkinghub.elsevier.com/retrieve/pii/S0377027303003020, 2004.

Schimmel, A. and Hübl, J.: Automatic detection of debris flows and debris floods based on a combination of infrasound and seismic signals, Landslides, 13, 1181–1196, https://doi.org/10.1007/s10346-015-0640-z, http://dx.doi.org/10.1007/s10346-015-0640-zhttp://link.springer.com/10.1007/s10346-015-0640-z, 2016.

Schneider, D., Bartelt, P., Caplan-Auerbach, J., Christen, M., Huggel, C., and McArdell, B. W.: Insights into rock-ice avalanche dynamics by combined analysis of seismic recordings and a numerical avalanche model, J. Geophys. Res. Earth Surf., 115, 1–20, https://doi.org/10.1029/2010JF001734, 2010.

Schneider, D., Huggel, C., Haeberli, W., and Kaitna, R.: Unraveling driving factors for large rock-ice avalanche mobility, Earth Surf. Processes, 36, 1948–1966, https://doi.org/10.1002/esp.2218, http://doi.wiley.com/10.1002/esp.2218, 2011.

Copyright waived. CC0 1.0.

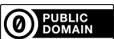



Schöpa, A., Chao, W.-A., Lipovsky, B. P., Hovius, N., White, R. S., Green, R. G., and Turowski, J. M.: Dynamics of the Askja caldera July 2014 landslide, Iceland, from seismic signal analysis: precursor, motion and aftermath, Earth Surf. Dynam., 6, 467–485, https://doi.org/10.5194/esurf-6-467-2018, https://www.earth-surf-dynam.net/6/467/2018/, 2018.

Schwaiger, H. F., Iezzi, A. M., and Fee, D.: AVO-G2S: A modified, open-source Ground-to-Space atmospheric specification for infrasound modeling, Comput. Geosci., 125, 90–97, https://doi.org/10.1016/j.cageo.2018.12.013, https://linkinghub.elsevier.com/retrieve/pii/S0098300418301560, 2019.

Tape, C., Christensen, D., Moore-Driskell, M. M., Sweet, J., and Smith, K.: Southern Alaska Lithosphere and Mantle Observation Network (SALMON): A Seismic Experiment Covering the Active Arc by Road, Boat, Plane, and Helicopter, Seismol. Res. Lett., 88, 1185–1202,

https://doi.org/10.1785/0220160229, https://pubs.geoscienceworld.org/srl/article/88/4/1185-1202/354115, 2017.

Uieda, L. and Wessel, P.: PyGMT: Accessing the Generic Mapping Tools from Python, in: AGU Fall Meet. 2019, pp. NS21B–0813, https://www.pygmt.org/, 2019.

Ulivieri, G., Marchetti, E., Ripepe, M., Chiambretti, I., De Rosa, G., and Segor, V.: Monitoring snow avalanches in Northwestern Italian Alps using an infrasound array, Cold Reg. Sci. Technol., 69, 177–183, https://doi.org/10.1016/j.coldregions.2011.09.006, http://dx.doi.org/10.

1016/j.coldregions.2011.09.006https://linkinghub.elsevier.com/retrieve/pii/S0165232X11001881, 2011.

Voight, B., ed.: Rockslides and Avalanches, 1: Natural Phenomena, vol. 14 of *Developments in Geotechnical Engineering*, Elsevier, 1978.

Waxler, R. M., Assink, J. D., Hetzer, C., and Velea, D.: NCPAprop—A software package for infrasound propagation modeling, J. Acoust. Soc. Am., 141, 3627–3627, https://doi.org/10.1121/1.4987797, http://asa.scitation.org/doi/10.1121/1.4987797, 2017.

Waythomas, C., Miller, T., and Begét, J.: Record of late Holocene debris avalanches and lahars at Iliamna Volcano, Alaska, J. Volcanol. Geoth.

Res., 104, 97–130, https://doi.org/10.1016/S0377-0273(00)00202-X, https://linkinghub.elsevier.com/retrieve/pii/S037702730000202X, 2000.

Waythomas, C. F. and Miller, T. P.: Preliminary volcano-hazard assessment for Iliamna Volcano, Alaska: U.S. Geological Survey Open-File Report 99-373, 31 p., https://pubs.usgs.gov/of/1999/0373/, 1999.

Werner, C. A., Doukas, M. P., and Kelly, P. J.: Gas emissions from failed and actual eruptions from Cook Inlet Volcanoes, Alaska, 1989–2006,

B. Volcanol., 73, 155–173, https://doi.org/10.1007/s00445-011-0453-4, http://link.springer.com/10.1007/s00445-011-0453-4, 2011.

Wessel, P., Luis, J. F., Uieda, L., Scharroo, R., Wobbe, F., Smith, W. H. F., and Tian, D.: The Generic Mapping Tools Version 6, Geochemistry, Geophys. Geosystems, 20, 5556–5564, https://doi.org/10.1029/2019GC008515, https://onlinelibrary.wiley.com/doi/abs/10.1029/2019GC008515, 2019.

Yamasato, H.: Quantitative Analysis of Pyroclastic Flows Using Infrasonic and Seismic Data at Unzen Volcano, Japan, J. Phys. Earth, 45,

397–416, https://doi.org/10.4294/jpe1952.45.397, http://joi.jlc.jst.go.jp/JST.Journalarchive/jpe1952/45.397?from=CrossRef, 1997.

Zimmer, V. L. and Sitar, N.: Detection and location of rock falls using seismic and infrasound sensors, Eng. Geol., 193, 49–60, https://doi.org/10.1016/j.enggeo.2015.04.007, http://dx.doi.org/10.1016/j.enggeo.2015.04.007https://linkinghub.elsevier.com/retrieve/pii/S0013795215001234, 2015.

Zimmer, V. L., Collins, B. D., Stock, G. M., and Sitar, N.: Rock fall dynamics and deposition: An integrated analysis of the 2009 Ahwiyah

Point rock fall, Yosemite National Park, USA, Earth Surf. Processes, 37, 680–691, https://doi.org/10.1002/esp.3206, 2012.