# Peer review of "Reconstructing the dynamics of the highly-similar May 2016 and June 2019 Iliamna Volcano, Alaska ice–rock avalanches from seismoacoustic data"

_Earth Surface Dynamics, 2020_

## Referee Comment (RC1) · Anonymous Referee #1 · 29 Jul 2020

The authors present here a double-case study of very similar ice-rock avalanches on the Iliamna Volcano, Alaska. The comparison of these two cases allows them to validate the inversion method they apply to the generated seismic waveforms to recover the force history of the avalanches. From the force history, they then compute their acceleration, velocity and directionality. Their inversion method is based on the hypothesis of the source being a point-source, spatially static. The main interest of this paper is that it reports two new case studies of avalanches, and that it compares the dynamics reconstructed using seismic signals with visual data (aerial photos, satellite imagery, and elevation data). The authors are giving a nice description of the data. I do not have the expertise to judge the details of the methods used, however it seems

to me that the methods are well-described and clear. Consequently, I recommend this paper for publication after some moderate revisions.

I have a few minor and moderate comments that I am stating here.

1) The authors should state more clearly what is the novelty of their study, especially in the introduction. I think that it would increase the impact of the paper. In the current shape, it is not clear what this study brings, compared to the references cited in the Introduction and Background sections.

2) In the abstract, the authors are stating: "Seismic and acoustic signals from these often-remote processes, combined with other geophysical observations, can provide key information for monitoring and rapid response efforts and enhance our understanding of event dynamics". I was expecting more discussion on this point in the main body of the paper. What is this study bringing regarding to this statement?

3) In general, the authors are having a nice discussion on their results, comparing them to other studies, discussing the limitations of the methods they are using. However, I think the paper would be improved by having a more fundamental discussion: what are these results telling us on the events, how can we use them to monitor this kind of events? Maybe the discrepancies between the inversion results for the two very similar events could be more discussed as well. (among other possible discussions)

4) Finally, it is not clear to me what is the use of the acoustic data in this study: the main results on the dynamics of the avalanches come from the seismic data. The acoustic data are occupying a large part in the title and main body. However, considering the output from this data, I would reduce their description, or emphasize better why they are new and important in this study.

Minor points:

1) The Introduction and Background sections are a bit long. They may be grouped?

2) In the legend of Figure 1: add the distances of the 2 closest stations

3) Line 70: Is there a reference?

4) Lines 188-189 : "The events also produced prodigious long-period energy with a dominant period of 35 s (Fig. 5)" What can be the source of this?

5) Figure 6, acoustic transmission loss: the patterns are pretty different, whereas the authors are stating that the sources are very similar. What can explain this discrepancy? (overall on the western part) I thought it could be due to the addition of acoustic stations in the western region, but these stations did not seem to detect any signal. I would like some discussion on this point.

6) Section 4.1.2, what is the definition of the root mean square pressure?

7) Lines 344-346: "We use the satellite imagery shown in Fig. 2 to estimate the mass for each event. First, we subtract the avalanche source area from the total area, ignore entrainment, and assume a uniform 1.5 m deposit thickness everywhere on the slope to obtain a volume." Is it not possible to deduce it from the DEMs?

8) Lines 356-368: It is not clear to me how the authors choose the end point.

9) Figure 8: Seismic and acoustic signals are shifted to be aligned on the time 0 of the inversion. But I do not understand why they are shifted for travel time from different points? (point force location for the seismic signals, and avalanche path midpoint for the acoustic signals?) Can the authors explain this choice, since it has an impact on the interpretation (paragraph beginning Line 505)?

10) Line 456: "manifested as a high-frequency": Indicate the frequency here. (Same Line 460)

---

## Referee Comment (RC2) · Velio Coviello (Referee) · 30 Jul 2020

The paper by Toney et al. presents the analysis of seismo-acoustic data of two ice–rock avalanches that occurred on Iliamna Volcano, Alaska (USA) on 2016 and 2019. The paper is well written and reports a unique dataset of seismic and infrasound observations of large mass wasting events. The methods employed are not new but their application to two recent events makes the paper definitely relevant for the community dealing with natural hazards in mountainous areas. I would recommend publication of this paper after minor revisions, here following my main comments to the authors.

1) Structure: the background and methods sections are quite long and have many sub-

sections. I would try to shorten the paper and simplify its structure. For instance, I would skip section 2.3 and move part of the text describing ice-rock avalanche to the Introduction. In addition, I suggest moving the text contained in the section 2.5 to the beginning of Data. Finally, I would skip the whole section 4.2.3 and move it all to a supplement file.

2) Dataset: did you consider extending your analysis to similar events that occurred at Iliamna Volcano before 2016? In Caplan-Auerbach and Huggel (2007) quite a lot of ice-rock avalanches are reported that produced seismic signals at Iliamna Volcano.

3) Methods: the inversion of low-frequency seismic data used to reconstruct the force history of the two ice–rock avalanches is a consolidated method. Given the large number of events (see point 1) and broadband seismic stations available, it would be possible to show and discuss the impact of the network geometry on the force history?

4) Event volumes: how did you estimate the value of 1.5 m deposit thickness? I would add the range of error to the event volumes. Ice-rock partition: is fifty-fifty consistent with field-based estimates of previous events? In any case, I do not expect that such an information on the volume uncertainty would explain the discrepancies between the masses inferred from the force inversion trajectories versus the ones calculated with satellite imagery. I suggest indicating where and when fragmentation and erosion-deposition processes occur, maybe adding some graphical features to figure 10 or some text to the description of stages A-E in section 6.2.

5) Results: quantitative results descend from the analysis of the seismic information. I appreciated the explicit acknowledgement of the limitations precluding the authors from assessing a complete infrasound source estimate. Actually, infrasound data are mainly used in the discussion to highlight the limitations of the force-history in describing the mass movements. However, I have the impression that section 6.4 can be extended mentioning that the transition from a block-type failure to a granular flow likely results in a higher frequency seismicity. Near-field seismoacoustic observations of debris flows

can support this discussion, see Hürlimann et al. (2019) and references therein.

6) Stick-slip activity: although this is not the objective of the paper, this is an intriguing point. I am wondering if precursory tremors like those mentioned in the paper can be produced by small ice-rockfall events preceding the main collapse. Progressive rockfall activity is a common process during the first phase of motion of a large landslide. What do you think?

References

Caplan-Auerbach, J., and Huggel, C., 2007, Precursory seismicity associated with frequent, large ice avalanches on Iliamna volcano, Alaska, USA: Journal of Glaciology, v. 53, p. 128–140, doi:10.3189/172756507781833866.

Hürlimann, M., Coviello, V., Bel, C., Guo, X., Berti, M., Graf, C., Hübl, J., Miyata, S., Smith, J.B., and Yin, H.Y., 2019, Debris-flow monitoring and warning: Review and examples: Earth-Science Reviews, v. 199, p. 102981, doi:10.1016/j.earscirev.2019.102981.

---

## Author Comment (AC3) · 9 Nov 2020

The discussion preprint has been very minorly edited to conform to U.S. Geological Survey standards. These small edits pertain to grammar, capitalization, and references and have no impact on the science of the paper. Because of these edits, some line numbers referenced in reviewer comments may be offset by one or two lines.

---

## Author Response (AR1)

Dear Referees and Associate Editor,

Thank you for your review of this manuscript. In this response file, please find our point-by-point responses to referee comments. The left column is the comment, and the right column is our response, with associated changes indicated if applicable. Locations of changes reflect the revised manuscript. (Please note that these point-by-point responses are also included as supplemental files in our Author Comments.)

The revised manuscript with changes highlighted follows our responses. Please note that the manuscript edits also reflect the input of a USGS internal reviewer, Mark Reid, who provided mostly minor comments.

Thank you very much for your time and effort.

Cheers,

Liam (on behalf of all co-authors)

**ESurf reviewer #1 — [anonymous]**

| Line | Reviewer comment | Our response |
|---|---|---|
| N/A | The authors should state more clearly what is the novelty of their study, especially in the introduction. I think that it would increase the impact of the paper. In the current shape, it is not clear what this study brings, compared to the references cited in the Introduction and Background sections. | A large component of the novelty of this study is the dataset — these are two very large events with very similar characteristics, *and* they were recorded extensively by acoustic and seismic sensors. We emphasize this in paragraph 3 of the Introduction section.
  We also stress the similarity of the events and accompanying benefits in the title, Abstract, and the first paragraph of Conclusions. |
| N/A | In the abstract, the authors are stating: "Seismic and acoustic signals from these often-remote processes, combined with other geophysical observations, can provide key information for monitoring and rapid response efforts and enhance our understand- ing of event dynamics". I was expecting more discussion on this point in the main body of the paper. What is this study bringing regarding to this statement? | We have added an additional section, §6.7, which explicitly addresses the feasibility of the force inversion method for rapid hazard response. |
| N/A | In general, the authors are having a nice discussion on their results, comparing them to other studies, discussing the limitations of the methods they are using. However, I think the paper would be improved by having a more fundamental discussion: what are these results telling us on the events, how can we use them to monitor this kind of events? Maybe the discrepancies between the inversion results for the two very similar events could be more discussed as well. (among other possible discussions) | Please see the above response to address your monitoring point.
  We hesitate to analyze the smaller-scale discrepancies between the two events, since these are more likely to arise from event-specific noise or station coverage factors (e.g., see the difference in vertical scaling between the 2016 and 2019 vertical trajectories).
  However — we have made a new section, §6.6, which incorporates an existing paragraph and now also discusses the observed difference in force-time function amplitudes between the two events. |
| N/A | Finally, it is not clear to me what is the use of the acoustic data in this study: the main results on the dynamics of the avalanches come from the seismic data. The acoustic data are occupying a large part in the title and main body. However, considering the output from this data, I would reduce their description, or emphasize better why they are new and important in this study. | We believe that the acoustic data reported on in this study validate their appearance in the title and main body.
  Firstly, acoustic recordings of large mass movements are rare; to have data from two highly similar events is even more so. We feel that it is important to present these data.
  Secondly, the acoustic data do provide complementary information to the seismic data. For instance, when both waveforms are aligned (Fig. 8) the absence of an initial acoustic transient to match the initial seismic transient suggests a sub- or inter-glacial source (as mentioned in the text) rather than a surface source such as a precursory rock or ice fall, since the latter would produce infrasound as well. Additionally, similar alignment of seismic and infrasound waveforms has been observed for other processes such as debris flows. Their alignment here may indicate something about the flow regime after fragmentation (stage B onwards).
  We've added additional discussion in §6.4. |

| N/A | The Introduction and Background sections are a bit long. They may be grouped? | We have removed a paragraph from the Introduction and moved one data-related section from the Background to Data, which makes both of those sections a more manageable length. |
|---|---|---|
| Fig. 1 | In the legend of Figure 1: add the distances of the 2 closest stations | The distances for these two stations now appear in the map. |
| 70 | Is there a reference? | Two sentences later, we provide several references. |
| 188–189 | "The events also produced prodigious long-period energy with a dominant period of 35 s (Fig. 5)" What can be the source of this? | The source of the long-period seismic radiation is discussed in the first paragraph of §2.1. |
| Fig. 6 | Figure 6, acoustic transmission loss: the patterns are pretty different, whereas the authors are stating that the sources are very similar. What can explain this discrepancy? (overall on the western part) I thought it could be due to the addition of acoustic stations in the western region, but these stations did not seem to detect any signal. I would like some discussion on this point. | In this response we assume that the patterns you refer to are the transmission loss patterns. Since the regional infrasound arrays all detected the signal, we focus on single infrasound sensors in the below discussion (inverted triangles and squares in Fig. 6). Our ability to detect an infrasound signal at a single infrasound sensor is strongly controlled by three factors:
1. The noise level at the sensor (indicated by sensor RMS pressure). Many of the "single-station" type infrasound sensors used in this study are part of the meteorological sensing package added to Transportable Array seismic stations. This means that the stations were sited primarily for seismic, not infrasonic, performance. Therefore, noise in the infrasonic band — for example, turbulence created by the interaction of wind with nearby topography (or trees/rocks/structures near the sensor) — can be large for these stations. Even for dedicated single sensor infrasound installs, such as those deployed for volcano monitoring, noise is an issue. Arrays can help mitigate this problem by determining coherent energy across the array.
2. The propagation conditions (indicated by transmission loss). For non-local (> 15 km) infrasound, propagation is especially important since entire portions of Earth's surface can reside in "shadow zones" in between bounce points of the atmospheric waveguides (ducts, see §5.1.3).
3. Source strength. We assume that the source strength and directionality are very similar between the two events, based upon the similarity of the acoustic data and deposits.
Fig. 6 is designed to display the first two of these factors on one map. In the case of the additional stations to the west in 2019, while we were better able to sample the wavefield, most of the added stations were noisy, so even with seemingly favorable propagation to the west, we have no additional detections. §5.1.2 discusses this. Finally, note that the transmission loss modeling is for a |

| | | basic point source and is independent of source character between the two years — variability of the transmission loss pattern corresponds to variability in the atmosphere between the two years. Likewise for the RMS pressure calculation — this solely reflects differences in local site noise between the two years. |
|---|---|---|
| Sec. 4.1.2 | what is the definition of the root mean square pressure? | For any time series signal, the root-mean-square is the square root of the mean of the squares of each data value. It is a relatively robust method for determining the average value — in this case, average pressure — of a waveform. |
| 344–346 | "We use the satellite imagery shown in Fig. 2 to estimate the mass for each event. First, we subtract the avalanche source area from the total area, ignore entrainment, and assume a uniform 1.5 m deposit thickness everywhere on the slope to obtain a volume." Is it not possible to deduce it from the DEMs? | It was not possible to deduce the avalanche volumes from DEM analysis. The SfM DEM was acquired in late July 2019, one month after the June 2019 event. Since the avalanche deposits have a large ice component, it is unlikely that the SfM DEM accurately captures the June 2019 event's deposit. More critically, we do not have pre-event DEMs for either event (nor a post-event DEM for the May 2016 event), precluding DEM subtraction.  We've added a sentence explaining this limitation to the new "Mass estimation" section (§4.1) of the manuscript. |
| 356–368 | It is not clear to me how the authors choose the end point. | We do not in fact pick a COM end point. We pick a COM start point, and find the COM runout length that results in the best match of the force-time function features to the topography and flow features evident in the deposits. We've added some clarification to this end in §4.3.3. |
| Fig. 8 | Seismic and acoustic signals are shifted to be aligned on the time 0 of the inversion. But I do not understand why they are shifted for travel time from different points? (point force location for the seismic signals, and avalanche path midpoint for the acoustic signals?) Can the authors explain this choice, since it has an impact on the interpretation (paragraph beginning Line 505)? | The selection of a source location to facilitate source-to-receiver distance calculations (and thus travel time removal) is difficult due to the moving source (COMs moved up to 8 km). We selected the avalanche path midpoint for the infrasound source location since this is likely the most acoustically energetic portion of the flow (see second and third paragraphs in §6.4 for discussion). The selection of a source location for the high-frequency (HF) seismic source is trickier, since we have identified multiple HF transients that correspond to different source locations. For example, the initial HF transient is associated with a failure near the crown of the avalanche, but the following spindle is associated with the fragmentation of mass further downslope (similar to the infrasound source location).  For consistency with the infrasound travel time removal location, we've changed the location for shifting of the seismic signals to the avalanche midpoint. This does not change our interpretations (paragraph beginning on line 509 in the revised manuscript). |
| 456, | "manifested as a high-frequency": Indicate the | Added "(> 5 Hz)" in these two places. See also Fig. |

| 460 | frequency here. (Same Line 460) | 4. |

**ESurf reviewer #2 — Velio Coviello**

| Line | Reviewer comment | Our response |
|------|-----------------|--------------|
| N/A | Structure: the background and methods sections are quite long and have many sub-sections. I would try to shorten the paper and simplify its structure. For instance, I would skip section 2.3 and move part of the text describing ice-rock avalanche to the Introduction. In addition, I suggest moving the text contained in the section 2.5 to the beginning of Data. Finally, I would skip the whole section 4.2.3 and move it all to a supplement file. | We have moved the text in §2.5 to the beginning of the Data section. We would like to retain §2.3 to provide some general context to the following description of Iliamna's ice–rock avalanches. We have moved the former §4.2.3 into a supplemental file. |
| N/A | Dataset: did you consider extending your analysis to similar events that occurred at Iliamna Volcano before 2016? In Caplan-Auerbach and Huggel (2007) quite a lot of ice-rock avalanches are reported that produced seismic signals at Iliamna Volcano. | It is true that there are many candidate Red Glacier avalanches which we could analyze. However, here we focused only on the two most recent events because: 1. These two events occurred during/after the deployment of the TA network, which provides us with acoustic as well as seismic data. We don't have acoustic recordings for earlier events. 2. There were very few broadband seismometers available in 2004 and earlier. An IRIS station query for stations within 200 km of Iliamna that were operational/available on 1 January 2006 reveals only 4 broadband stations available — 3 Güralp 6TD's (30 s corner) and 1 Güralp 40T (60 s corner). The 2016 inversion also benefited from the presence of the temporary SALMON seismic array. 3. Availability of auxiliary data such as satellite and ground-based imagery is not as good for the events from 2004 and earlier. |
| N/A | Methods: the inversion of low-frequency seismic data used to reconstruct the force history of the two ice–rock avalanches is a consolidated method. Given the large number of events (see point 1) and broadband seismic stations available, it would be possible to show and discuss the impact of the network geometry on the force history? | Though there are a relatively large number of Red Glacier avalanches, only these two events have sufficient data for inversion (see response immediately above). Therefore, the best way to explore the sensitivity of inversion results to changing network geometry is via synthetic examples. We do not perform a formal investigation of this effect here, but we plan to address the network geometry consideration in an upcoming force inversion "best practices" paper. The jackknifed trajectories shown in this paper serve to convey some idea of the expected spread of the solutions under changing input data. But we lack a high-quality balance of three component data (most data used here is vertical) to understand this issue from real data alone — hence the need for synthetic testing. |
| N/A | Event volumes: how did you estimate the value of 1.5 m deposit thickness? I would add the range of error to the event volumes. Ice-rock partition: is | The deposits for these two events were not measured directly, so we must make an educated guess. Previous studies have been forced to do the |

| | fifty-fifty consistent with field-based estimates of previous events? In any case, I do not expect that such an information on the volume uncertainty would explain the discrepancies between the masses inferred from the force inversion trajectories versus the ones calculated with satellite imagery. I suggest indicating where and when fragmentation and erosion- deposition processes occur, maybe adding some graphical features to figure 10 or some text to the description of stages A-E in section 6.2. | same — Huggel et al. (2007) estimated deposit thicknesses of 1–3 m for a 2004 avalanche on Iliamna's Lateral Glacier. Waythomas et al. (2000) estimated 1.5 m thickness and a composition of "at least 50 percent" ice/snow for 1996 and 1997 Red Glacier avalanches. We've added a citation for the 50–50 composition, and attached upper and lower bounds to the 1.5 m thickness estimate which we propagate into the volume and mass calculations.

We can't make too many statements about erosion/deposition processes based upon our limited groundtruth. We now mention fragmentation in both stages B and C. |
|---|---|---|
| N/A | Results: quantitative results descend from the analysis of the seismic information. I appreciated the explicit acknowledgement of the limitations precluding the authors from assessing a complete infrasound source estimate. Actually, infrasound data are mainly used in the discussion to highlight the limitations of the force-history in describing the mass movements. However, I have the impression that section 6.4 can be extended mentioning that the transition from a block-type failure to a granular flow likely results in a higher frequency seismicity. Near-field seismoacoustic observations of debris flows can support this discussion, see Hürlimann et al. (2019) and references therein.

Hürlimann, M., Coviello, V., Bel, C., Guo, X., Berti, M., Graf, C., Hübl, J., Miyata, S., Smith, J.B., and Yin, H.Y., 2019, Debris-flow monitoring and warning: Review and examples: Earth-Science Reviews, v. 199, p. 102981, doi:10.1016/j.earscirev.2019.102981. | Thank you for alerting us to this very relevant review paper. We've added some discussion in §6.4 regarding the similarity of high-frequency seismic and infrasound waveforms for the 2016 and 2019 events and the resemblance of this observation to some of the debris flow studies referenced in Hürlimann et al. (2019). This provides additional evidence that Iliamna ice–rock avalanches transition into a granular flow that (at least seismoacoustically) exhibits flow dynamics similar to those of debris flows. |
| N/A | Stick-slip activity: although this is not the objective of the paper, this is an intriguing point. I am wondering if precursory tremors like those mentioned in the paper can be produced by small ice-rockfall events preceding the main collapse. Progressive rockfall activity is a common process during the first phase of motion of a large landslide. What do you think? | Caplan-Auerbach and Huggel (2007) make a compelling case for the origin of the Iliamna avalanche precursory signals being on the ice-rock interface or within ice. The precursory, transient signals are found to be highly similar and their inter-event timing shrinks as the time to failure approaches. See e.g. Fig. 10 in Caplan-Auerbach and Huggel (2007). While precursory ice-rockfall activity may certainly be intermittently present, the high similarity of the precursory transient signals and their reliable increase in event frequency is more consistent with a sub- or intra-glacial source. |

[revised manuscript text omitted]
_{\mathrm{Z}}(t) = [f_{\mathrm{N}}(t)\cos\phi + f_{\mathrm{E}}(t)\sin\phi] * g_{\mathrm{ZH}}(t) + f_{\mathrm{Z}}(t) * g_{\mathrm{ZV}}(t)\,, \tag{1}$$

$$u_{\mathrm{R}}(t) = [f_{\mathrm{N}}(t)\cos\phi + f_{\mathrm{E}}(t)\sin\phi] * g_{\mathrm{RH}}(t) + f_{\mathrm{Z}}(t) * g_{\mathrm{RV}}(t)\,, \text{ and} \tag{2}$$

300 $\quad u_{\mathrm{T}}(t) = [f_{\mathrm{N}}(t)\sin\phi - f_{\mathrm{E}}(t)\cos\phi] * g_{\mathrm{TH}}(t)\,, \tag{3}$

where the symbol $*$ denotes convolution (Herrmann, 2013). $\boldsymbol{f}(t) = [f_{\mathrm{Z}}(t),\ f_{\mathrm{N}}(t),\ f_{\mathrm{E}}(t)]$ is the 3D force-time function exerted on the Earth by the avalanche in terms of vertical (Z), north (N), and east (E) components and $\phi$ is the source-to-station azimuth measured clockwise from north. The Green's functions $g_{\mathrm{ZV}}(t)$, $g_{\mathrm{ZH}}(t)$, $g_{\mathrm{RV}}(t)$, $g_{\mathrm{RH}}(t)$, and $g_{\mathrm{TH}}(t)$  describe how vertical (Z), radial (R), and transverse (T) components of displacement are excited by vertical (V)

305 and horizontal (H) force impulses. ~~Note that in the remainder of the text we use $\boldsymbol{f}$ when discussing the force-time function as a model vector – that is, a 1D column vector consisting of the three components of the force-time function concatenated end-to-end. We use $\boldsymbol{f}(t)$ when referring to the 3D force-time function. These two symbols represent the same object. We use the same convention for $\boldsymbol{u}$ and $\boldsymbol{u}(t)$.~~

**4.3.3**

310 ~~In numerical contexts, it is more convenient to formulate the convolution as a matrix multiplication. We therefore transform the GFs into convolution matrices $\boldsymbol{\Lambda}$ by reversing the GFs in time and staggering them as in Allstadt (2013), where the time dependence of the GF is now implicitly stored in the matrix. (For example, the multiplication $\boldsymbol{\Lambda}_{\mathrm{ZV}}\boldsymbol{f}_{\mathrm{Z}}$ corresponds to the convolution $f_{\mathrm{Z}}(t) * g_{\mathrm{ZV}}(t)$; see Allstadt (2013), Eq. A5.) Making this modification, we can combine Eqs. 1–3 into~~

$$\underline{\boldsymbol{u}^k = \boldsymbol{\Gamma}^k \boldsymbol{f}}\,,$$

315

$$\boldsymbol{\Gamma}^k = \begin{bmatrix} \boldsymbol{\Lambda}^k_{\mathrm{ZV}} & \boldsymbol{\Lambda}^k_{\mathrm{ZH}}\cos\phi^k & \boldsymbol{\Lambda}^k_{\mathrm{ZH}}\sin\phi^k \\ \boldsymbol{\Lambda}^k_{\mathrm{RV}} & \boldsymbol{\Lambda}^k_{\mathrm{RH}}\cos\phi^k & \boldsymbol{\Lambda}^k_{\mathrm{RH}}\sin\phi^k \\ \boldsymbol{0} & \boldsymbol{\Lambda}^k_{\mathrm{TH}}\sin\phi^k & -\boldsymbol{\Lambda}^k_{\mathrm{TH}}\cos\phi^k \end{bmatrix}.$$

$$\underline{\boldsymbol{d} = \mathbf{G}\boldsymbol{f}}\,,$$

320  We invert for $\underline{\boldsymbol{f}(t)}$ using a higher-order Tikhonov-regularized

$$\underline{\boldsymbol{f} = \left[\mathbf{G}^\top\mathbf{G} + \alpha^2\left(a_0\mathbf{I} + a_1\mathbf{L}_1{}^\top\mathbf{L}_1 + a_2\mathbf{L}_2{}^\top\mathbf{L}_2\right)\right]^{-1}\mathbf{G}^\top\boldsymbol{d}}\,,$$

where $\mathbf{I}$ is the identity matrix and $\mathbf{L}_1$ and $\mathbf{L}_2$ are first- and second-order roughening matrices which approximate the first and second derivatives, respectively. The coefficients $a_0$, $a_1$, and $a_2$ control the degree of importance given to "small," "flat," and "smooth" models, respectively. They must sum to one:

$$\sum_{i=0}^{2} a_i = 1 .$$

The regularization parameter $\alpha$ is chosen to balance the constraints on the model specified by the $a_i$ coefficients while still fitting the data well. We use the L-curve criterion (Hansen, 1992) to find the optimal value for $\alpha$. For both inversions we found the optimal values for these parameters were $\alpha = 4.8 \times 10^{-20}$ and $a_i = [0.4, 0, 0.6]$. This selection of $a_i$'s prioritizes a model that is both small in magnitude (more centered on zero) and smooth. The inclusion of the higher-order regularization matrices $\mathbf{L}_1$ and $\mathbf{L}_2$ in Eq. A4 separates this method from the method used in Allstadt (2013) and (Coe et al., 2016), which only included zeroth-order Tikhonov regularization. approach which we describe in detail in Appendix A.

To characterize the fit of the model to the data, we compute the variance reduction (VR), which is defined as

$$\mathrm{VR} = \left( 1 - \frac{\|\boldsymbol{d} - \boldsymbol{d}_{\mathbf{obs}}\|^2}{\|\boldsymbol{d}_{\mathbf{obs}}\|^2} \right) \times 100\% ,$$

where $\boldsymbol{d}_{\mathbf{obs}}$ are the observed data and $\boldsymbol{d}$ are the synthetic data predicted by the forward model (Eq. A3).

In addition to regularization, we constrain all of the components of $\boldsymbol{f}(t)$ to sum to zero to conserve the total momentum of the Earth (see Allstadt, 2013, Appendix A). We also enforce all components of $\boldsymbol{f}(t)$ be zero prior to a specified "zero time." We choose the zero time to correspond to the point where the vertical component $f_{\mathrm{Z}}(t)$ is non-zero and rising, signaling the initial downward acceleration of the avalanche. The zero time for the 22 May 2016 event is 07:57:53 and the zero time for the 21 June 2019 event is 00:03:08. The selection of the zero time was unambiguous for both events.

To assess the stability of the inversion, we use the jackknife technique (e.g., Moretti et al., 2015; Coe et al., 2016). We run 20 iterations of the inversion, each time randomly discarding 30% of the waveforms.

**4.3.3 Trajectory calculations**

For simple mass movements, the trajectory can be calculated from the force-time function if the mass is known or can be estimated. The acceleration felt by the avalanche COM is given by Newton's second law

$$\boldsymbol{a}(t) = -\frac{\boldsymbol{f}(t)}{m} , \tag{4}$$

where $m$ and $\boldsymbol{a}(t)$ are the mass and acceleration of the avalanche, respectively. The sign change arises from the fact that $\boldsymbol{f}(t)$ is equal but opposite to the force felt by the avalanche. Integrating twice with respect to time yields the displacement. Since the avalanche paths are straightforward and we have two stable inversions, we apply the double integration method to obtain trajectories for the 2016 and 2019 events. Note that this method assumes that the mass $m$ is constant, which is clearly not the case due to entrainment and deposition along the path. We start integration at the zero time and end at 200 s since the forces are

essentially zero at this point. Unlike previous inversions, we add an additional, intuitive constraint that the velocity go to zero at the end of avalanche. This was implemented for each component of the velocity by subtracting a linear trend starting at zero

[revised manuscript text omitted]

calculations yield volumes of $(13 \pm 8)$ million m$^3$ in 2016 and $(11 \pm 7)$ million m$^3$ in 2019. The corresponding masses are $(22 \pm 14)$ billion kg in 2016 and $(19 \pm 13)$ billion kg in 2019.

**6   Discussion**

**6.1   Acoustic source directionality**

460   We lack sufficient infrasound station coverage to fully test an applicable acoustic source model, such as a single or distributed dipole, so we do not attempt an acoustic source inversion (e.g., Kim et al., 2012; Iezzi et al., 2019) here. Our infrasound analysis is therefore largely qualitative. By modeling infrasound propagation and site noise conditions, we sought to isolate source properties, such as size and directionality, from path effects. For example, Perttu et al. (2020) found that atmospheric propagation effects could not explain the infrasound radiation pattern observed for the 2018 Anak Krakatau flank collapse, and

465   used this to infer that the collapse acted like a piston, pushing sound in a directed manner.

For the Iliamna avalanches, examination of acoustic propagation alone might lead one to believe that source directionality is present, given the consistent detections of stations to the east of Iliamna despite variable local propagation conditions between the two events. However, there are two complicating factors in our case. First, station noise analysis (Fig. 6) shows that local stations to the west of Iliamna (and to the north and south as well in 2019) had high noise levels, indicating that

470   preferential detection on stations to the east could simply be due to lower noise levels at those stations. Second, while rugged topography surrounds Iliamna, there is less topography blocking propagation to the east than to the west (Fig. 1). Furthermore, the avalanches occurred on the east flank of Iliamna. Since infrasound propagation at local distances is strongly controlled by topography (Kim et al., 2015), propagation to the east from Iliamna may be topographically preferred. These complicating factors preclude us from assessing source directionality or obtaining quantitative source estimates.

**475   6.2   Multi-stage failure and flow**

Synthesis of the force-time function with high-frequency waveforms and force-derived COM acceleration magnitudes and speeds (Fig. 8) as well as force-derived trajectories (Fig. 10) suggests a consistent multi-stage failure and flow pattern for both avalanches. Our interpretation is as follows, with approximate times relative to the inversion zero time as well as color codes given in brackets:

480   A.  Initial failure of the source region in ice or at the ice–bedrock interface and subsequent sliding at an average angle of ∼20–25°, manifested as a high-frequency ($> 5$ Hz) seismic transient and a substantial eastward acceleration. No detectable infrasound is generated by  the initial failure (the small pulse visible at ∼10 s in 2016 is not seen on any other stations or arrays and is therefore likely noise) [0–20 s; red].

B.  The avalanche mass reaches its maximum speed and material becomes fragmented, changing the flow regime from
485   coherent to granular and turbulent. This is manifested as a gradual increase in the high-frequency ($> 5$ Hz) seismic energy; infrasound energy begins to rise simultaneously as the flow bends to the south [20–50 s; orange].

C. The now-fragmented flow bends to the north and then to the south as both high-frequency seismic and infrasound signals reach their peak amplitudes. Flow speeds decrease but stay between ~40–60 m s$^{-1}$ [50–90 s; light green]~30–60 m s$^{-1}$ [50–85 s; light green].

D. The flow encounters a change to a shallower slope angle (<$\leq$ 10°) where a tributary glacier joins Red Glacier from the southwest (see Fig. 2). This is manifested as an impulsive, relatively short-period (~30 s) downward force. The high-frequency seismic and infrasound signals taper off and flow speeds continue a slow decline [90–105 s; dark green][85–105 s; dark gre

E. After passing the kink in topography where slope angle decreases, the flow broadens and decelerates, forming the wide, flat debris lobe seen in Fig. 2. The east component of $\boldsymbol{f}(t)$ is largely positive, indicating deceleration of the flow. The vertical component of $\boldsymbol{f}(t)$ is near-zero, and this portion of the flow is not seismically or acoustically energetic for frequencies > 0.1 Hz $> 0.1$ Hz [105+ s; blue].

Our trajectories are compatible with numerical flow models for Red Glacier avalanches performed by Huggel et al. (2007) and Schneider et al. (2010), which both indicate that the avalanche COM tends to the orthographic orographic downslope right and then orthographic downslope left, in the latter case forming a superelevation-like flow lobe visible in Fig. 2. We do not consider the possibility that the observed deposit was formed by two separate flows, as suggested by Huggel et al. (2007) for the 1980 and 2003 Red Glacier avalanches, as we do not see evidence for two separate flows in the seismoacoustic signals or in satellite imagery of the deposits (Fig. 2) and our modeling assuming a single flow is compatible with previous modeling and observations. This suggests that only one flow took place, at least in 2016 and 2019.

**6.3 Mass estimation**

One complication of extracting quantitative information from the force inversion results concerns the method of regularization. Since we impose penalties on the size, slope, and roughness of $\boldsymbol{f}$ $\boldsymbol{f}(t)$ via the $a_i$ coefficients, the resultant force amplitudes are likely artificially depressed compared to the true values, as mentioned in §4.3.3. This is evidenced by the much smaller magnitude of the masses from the force inversion trajectories versus our satellite imagery based estimates (2.1 and 3.0 billion kg versus 22 and 19 billion kg for the 2016 and 2019 events, respectively). Even the lower bounds on our imagery-based mass estimates are still far larger than their inversion-derived equivalents, suggesting that the force amplitudes are indeed being suppressed by the regularization scheme (see Appendix A). Additionally, we are inverting a band-limited signal – energy present at very long periods (> 80 s $> 80$ s) is not reflected in $\boldsymbol{f}$ $\boldsymbol{f}(t)$, which also artificially depresses $\boldsymbol{f}$ $\boldsymbol{f}(t)$. Due to these biases, we do not apply the scaling relationship of Ekström and Stark (2013) to these results. We note that in general the masses of these events are not well constrained due to poor constraints on deposit thickness and the relative contributions of entrainment and deposition to the total failure mass. Better groundtruth estimates ground observations of avalanche deposit properties would help constrain the effect of regularization, and we encourage such studies in the future. We do note that the phase and *relative* amplitude of the force-time function between the two events (Fig Figs. 8a–f) are not affected by the regularization.

**6.4 Flow dynamics**

520 Average flow speeds for mass movements can be estimated from the duration of the high-frequency seismic envelope if the horizontal runout length $L$ is known (Caplan-Auerbach et al., 2004). However, it is often difficult to estimate the duration of the flow from the seismic envelope, since the noise floor can bury the earliest and latest parts of the emergent signal (Huggel et al., 2007). Caplan-Auerbach and Huggel (2007) estimated average flow speeds of about 20–50 m s$^{-1}$ for Red Glacier avalanches using this method. A more complete assessment of flow speeds can be obtained from numerical modeling or by examination

525 of the speed time series $\|\boldsymbol{v}(t)\|$ obtained from the force inversion,  as these provide both average and maximum values. Schneider et al. (2010) found an average flow speed of about 50 m s$^{-1}$ and peak flow speeds between 70–100 m s$^{-1}$ for a numerically modeled 2003 Red Glacier avalanche. Our values derived from $\|\boldsymbol{v}(t)\|$ (average speeds of 33 and 34 m s$^{-1}$ and maximum speeds of 75 and 74 m s$^{-1}$ in 2016 and 2019, respectively) are compatible with these results as well as those of Caplan-Auerbach and Huggel (2007), though we note that our values describe COM dynamics, while those of Caplan-Auerbach

530 and Huggel (2007) are calculated using $L$ and therefore apply to the flow front. Our results are also broadly compatible with other studies of similar large avalanches, such as the July 2007 Mount Steele rock–ice avalanche (35–65 m s$^{-1}$ average speed; Lipovsky et al., 2008) and the June 2016 Lamplugh Glacier rock avalanche ($\sim$55 m s$^{-1}$ maximum speed; Dufresne et al., 2019).

Inversion-derived COM acceleration magnitudes $\|\boldsymbol{a}(t)\|$ and flow speeds $\|\boldsymbol{v}(t)\|$ are plotted in Figs. 8g–j. The peak in-

535 frasound amplitude does not correlate with peak acceleration magnitude nor peak speed, instead occurring about 50 s after the latter. This notable latency between peak speed and peak acoustic energy might be explained by a model similar to Marchetti et al. (2019b), where infrasound is produced by waves at the free surface of the flow. Such waves would take time to develop since the initially blocky mass needs to be sufficiently fragmented and turbulent, which requires high flow speeds. The infrasound and seismic waveforms (Figs. 8k–n) do exhibit similar shapes and reach their peak values at approximately the

540 same time (after travel time removal). This alignment of high-frequency seismic and infrasound signals has previously been observed for debris flows (Schimmel et al., 2018; Marchetti et al., 2019b) and suggests that after some initial breakup period, Iliamna ice–rock avalanches may exhibit similar flow dynamics to debris flows, at least seismoacoustically.

Another possibility is that flow interaction with a particular topographic feature along the flow path is generating infrasound, and that the observed peak amplitude timing corresponds to the travel time for the flow to reach this feature. Figures

545 8m and n indicate that peak infrasound occurs anywhere from 50–85 s into the flow. Because the prominent northward force linked to the flow lobe on the orographically downslope left side of the flow occurs at about 80 s, flow turbulence at this point could be responsible for the peak in infrasound energy. Moore et al. (2017) observed a ground-coupled airwave associated with the second of two very large rock avalanches at Bingham Canyon Mine (Utah, USA). They inferred from the timing of the phase that the airwave was likely coupled into the ground when the rock

[revised manuscript text omitted]

595  One notable difference between the force-time functions obtained for the two avalanches is the increased amplitude for the 2019 event. This increase is consistent across all three force components, yet it is unlikely to be an inversion artifact since both inversions have the same regularization parameters. The high-frequency seismic signals (Figs. 8k and l) also indicate larger amplitudes for the 2019 event. Since the observed deposits have similar sizes (see Fig. 2), this suggests that a larger amount of mass was moved in 2019 than in 2016 but with little change in runout length. The mass discrepancy could be caused by
600  varying initial failure thicknesses (i.e., a thicker crown in 2019) or an increased portion of rock involved in the 2019 event versus the 2016 event. Unfortunately, we do not possess the field-based observations and measurements necessary to test these hypotheses.

**6.7 Feasibility for rapid hazard response**

The detailed information on avalanche dynamics retrievable from the rapidly-recorded seismic signals for these events raises
605  the question of the suitability of this method for near-real-time applications. Besides the seismic signals themselves, only two independent pieces of information are required to obtain 3D trajectories: The event location (for locating the point force) and the failure mass (for converting force to acceleration). In this study, we used high-resolution satellite imagery to estimate the location and to inform the selection of a mass.

In the absence of any independent data, the following could be performed: The event location could be estimated using
610  traditional earthquake or mass movement-specific location methods (see Allstadt et al., 2018, and references therein), and the failure mass could be roughly estimated from the scaling laws of Ekström and Stark (2013). A location could also be determined from infrasound signals using backprojection (see e.g. Sanderson et al., 2020). Note that due to the long wavelengths of the signals used, a precise location is not critical for the inversion process. The resulting seismically derived trajectory would be a rough approximation due to uncertainties in mass estimation and/or location. However, the *directionality* and relative size of
615  the mass movement would be preserved, and this information could be harnessed to remotely determine the likely path and scale of a mass movement.

Unfortunately, automatic locations are not available for the two Iliamna events or other events of comparable size. However, we note that the very large June 2016 Lamplugh Glacier, Alaska rock avalanche (see Bessette-Kirton et al., 2018; Dufresne et al., 2019)

has a cataloged location and origin time. In general, at this time an automatic inversion method would likely be successful only for very large mass movements with high SNR seismic and acoustic waveforms. We note that our methods are primarily aimed at providing information complementary to other techniques; they do not currently constitute a stand-alone or automated technique. Still, in remote settings where event information from other sources may be delayed or unavailable – such as Alaska – this approach could provide key estimates of basic flow properties in near-real-time.

**7 Conclusions**

Surficial mass movements transfer energy into the solid Earth and the atmosphere, producing seismoacoustic signals that yield complementary information about event dynamics. In this study, we  analyze an exceptional seismoacoustic dataset from two large, highly similar ice–rock avalanches to reconstruct the dynamics of the events. The  similarity of these avalanches provides an excellent opportunity to test the robustness of our modeling methods. Our force-time functions are derived from the inversion of long-period (15–80 s) seismic signals recorded on stations  $> 80$ km from the avalanches. They indicate that over the course of about 150 s the avalanche COM slid to the east, was subsequently deflected slightly to the south and then to the north, and then broadly decelerated. Our results provide constraints on time-varying avalanche acceleration, velocity, and directionality. This is important for hazard mitigation as well as general understanding of seismic signals from mass movements, though better estimates of mass and flow properties from field studies (e.g., Dufresne et al., 2019) and numerical modeling (e.g., Moretti et al., 2012) are needed to fully exploit this method's potential.

While it was possible to model the avalanche seismic source, we lacked sufficient infrasound data to quantitatively characterize the acoustic source. After accounting for propagation effects and station noise, we cannot assess whether the Iliamna avalanches exhibit acoustic source directionality. Still, the acoustic data are qualitatively consistent with our force-derived reconstructions. It appears that infrasound from these avalanches is produced after the mass movement regime transitions from cohesive block-type failure to granular and turbulent flow, but controlled experiments and denser acoustic instrumentation are needed to fully test this hypothesis.

Iliamna Volcano is an excellent  site for the seismoacoustic and geomorphological study of these impressive avalanches due to their relatively frequent occurrence at the volcano. Future work at Iliamna – as well as at other sites of repetitive surficial mass movements – should synthesize advanced numerical modeling techniques with detailed  observations including video footage and repeat high-resolution DEM acquisitions. These efforts, combined with more complete acoustic station coverage – perhaps with arrays as well as single sensors – could result in a substantial increase in our understanding of the behavior of large debris avalanches and other mass movements. This insight may then be applicable for mitigation of, and response to, the significant hazards posed by these  dramatic surface processes.

*Data availability.* All of the seismic and infrasound data used in this study are available from the Incorporated Research Institutions
for Seismology Data Management Center (IRIS DMC). The CPS model file we used to compute GFs for the inversions is available at
eas.slu.edu/eqc/eqc_cps/TUTORIAL/SPHERICITY/AK135/tak135sph.mod.

**Appendix A: Inversion formulation and constraints**

Consider the convolutions given by Eqs. 1–3. In numerical contexts, it is more convenient to formulate these convolutions as
matrix multiplications. We therefore transform the Green's functions (GFs) into convolution matrices $\mathbf{\Lambda}$ by reversing the GFs
in time and staggering them as in Allstadt (2013), where the time dependence of the GF is now implicitly stored in the matrix.
(For example, the multiplication $\mathbf{\Lambda_{ZV}} \boldsymbol{f_Z}$ corresponds to the convolution $f_Z(t) * g_{ZV}(t)$; see Allstadt (2013), Eq. A5.) Making
this modification, we can combine Eqs. 1–3 (dropping the explicit time dependence for brevity) into

$$\boldsymbol{u}^k = \mathbf{\Gamma}^k \boldsymbol{f}, \tag{A1}$$

where now the superscript $k$ denotes the station and $\mathbf{\Gamma}^k$ is a matrix of GF convolution matrices:

$$\mathbf{\Gamma}^k = \begin{bmatrix} \mathbf{\Lambda_{ZV}^k} & \mathbf{\Lambda_{ZH}^k} \cos\phi^k & \mathbf{\Lambda_{ZH}^k} \sin\phi^k \\ \mathbf{\Lambda_{RV}^k} & \mathbf{\Lambda_{RH}^k} \cos\phi^k & \mathbf{\Lambda_{RH}^k} \sin\phi^k \\ \mathbf{0} & \mathbf{\Lambda_{TH}^k} \sin\phi^k & -\mathbf{\Lambda_{TH}^k} \cos\phi^k \end{bmatrix}. \tag{A2}$$

We can now write the linear forward model for $N$ stations as

$$\boldsymbol{d} = \mathbf{G}\boldsymbol{f}, \tag{A3}$$

[revised manuscript text omitted]

We thank Hannah Dietterich and Tim Orr for their contributions to the Iliamna SfM project. Jacqueline Caplan-Auerbach and Chris Waythomas helped us understand the recent history and typical character of Iliamna avalanches. Taryn Lopez provided us with information

about zones of recent fumarolic activity on Iliamna. We thank Carl Tape for early access to embargoed SALMON data. The quality of this manuscript was significantly improved by the helpful comments of Mark Reid, Velio Coviello, and one anonymous reviewer.

R. Matoza was supported by NSF grants EAR-1614855 and EAR-1847736. The authors also acknowledge support from NSF grant EAR-1614323 and from AVO through the U.S. Geological Survey Volcano Hazards Program.

705    Ch'naqal'in (Iliamna Volcano) is located on the  ancestral *ełnana* (land) of the Dena'ina people of south-central Alaska.

[revised manuscript text omitted]

---

## Referee Report (RR1)

**Second Review of "Reconstructing the dynamics of the highly-similar May 2016 and June 2019 Iliamna Volcano, Alaska ice–rock avalanches from seismoacoustic data", Toney et al.**

The authors have nicely improved their manuscript, and addressed my previous comments. I therefore recommend this paper for publication.

I have technical and very minor comments that I am stating here. All the line numbers are referring to the revised manuscript including the track changes.

1) Line 184: Remove one of the two "was"

2) Paragraph 4.2.2: Is what you describe in this paragraph what is represented by the blue scale color in Fig.6? If yes, please make reference to the figure in this paragraph.

3) Lines 282-283: "This constraint ensures that we only use stations for which the source-receiver distance changed by a maximum amount of 10% over the course of the event."  You can also add here that this allows you to consider the source as a point source.

4) Lines 351-352: "Note that this method assumes that the mass m is constant, which is clearly not the case due to entrainment and deposition along the path."  So do you assume that variations in the mass are negligible and so allow you to use this method?

5) In paragraph 6.5, you say "In this study, we used high-resolution satellite imagery to estimate the location and to inform the selection of a mass." But from what I understand reading Line 377 "The trial mass starts at zero (giving an infinite length) and is increased in increments of 10 million kg until the length calculated with the trial mass drops below the target length." is that you did not use the masses computed from satellite imagery as a-priori for your inversion. I would remove " and to inform the selection of a mass." or detail this statement more.